# Relationships of stomatal morphology to the environment across plant communities

Congcong Liu[1,2,3], Lawren Sack [4], Ying Li[3], Jiahui Zhang[3], Kailiang Yu[5], Qiongyu Zhang[3], Nianpeng He [3,6,7] ✉ & Guirui Yu [3,8]

The relationship between stomatal traits and environmental drivers across plant communities has important implications for ecosystem carbon and water fluxes, but it has remained unclear. Here, we measure the stomatal morphology of 4492 species-site combinations in 340 vegetation plots across China and calculate their community-weighted values for mean, variance, skewness, and kurtosis. We demonstrate a trade-off between stomatal density and size at the community level. The community-weighted mean and variance of stomatal density are mainly associated with precipitation, while that of stomatal size is mainly associated with temperature, and the skewness and kurtosis of stomatal traits are less related to climatic and soil variables. Beyond mean climate variables, stomatal trait moments also vary with climatic seasonality and extreme conditions. Our findings extend the knowledge of stomatal trait–environment relationships to the ecosystem scale, with applications in predicting future water and carbon cycles.

Stomata are micropores in the leaf epidermis, bounded by a pair of guard cells that regulate the exchange of $CO_2$ and water vapor between the leaf and the atmosphere. The stomata have been influencing plant adaptation and functions ever since their evolution enabled plants to colonize land[1–3]. To optimize carbon fixation per unit water loss, plants can adjust stomatal pore aperture in the short term and modify stomatal traits such as stomatal density and size in the long term. A great number of studies have found that stomatal traits affect the drought resistance and water use efficiency of plants. Thus, stomatal traits are frequently and increasingly used in many fields of biology, including ecology and agriculture[4]. Across diverse species, a negative relationship between stomatal density and size has been well characterized[5]. By contrast, relatively few studies have focused on stomatal traits at the community level, although community-scale plant functional traits influence ecosystem functions[6–8]. Thus, analysis of community stomatal traits and their association with environmental variables at a large scale is important for resolving how climate change affects ecosystem functioning, including ecosystem productivity and water use efficiency[9,10].

A growing literature has revealed stomatal trait-environment relationships within and across species[11–13]. However, mechanisms by which environmental variation drives trait variation could fundamentally differ depending on the ecological scale[14], with trait plasticity and ecotypic adjustment influencing trait variation within species and macroevolutionary processes influencing variation across species. At the community level, trait variation is mainly determined by the ecological processes of community assembly. Therefore, whether conclusions drawn from within or across species can be applied at the community level is unclear. Previous studies suggested that species with diverse ecological strategies often co-exist within the same community[15,16], resulting in an inability to predict trait-environment relationships at the community level.

[1]Key Laboratory of Ecology and Environment in Minority Areas (Minzu University of China), National Ethnic Affairs Commission, 100081 Beijing, China. [2]College of Life and Environmental Sciences, Minzu University of China, 100081 Beijing, China. [3]Key Laboratory of Ecosystem Network Observation and Modeling, Institute of Geographic Sciences and Natural Resources Research, Chinese Academy of Sciences, 100101 Beijing, China. [4]Department of Ecology and Evolutionary Biology, University of California, Los Angeles, CA 90025, USA. [5]Department of Ecology and Evolutionary Biology, Princeton University, Princeton, New Jersey 08540, USA. [6]Center for Ecological Research, Northeast Forestry University, 150040 Harbin, China. [7]Earth Critical Zone and Flux Research Station of Xing'an Mountains, Chinese Academy of Sciences, 165200 Daxing'anling, China. [8]College of Resources and Environment, University of Chinese Academy of Sciences, 100049 Beijing, China. ✉e-mail: henp@igsnrr.ac.cn

There are two common approaches to exploring trait-environment relationships of plant communities at the regional or global scale. One is based on plant trait databases (such as TRY trait database, https://www.try-db.org/) and species occurrence databases, for example, Global Biodiversity Information Facility (GBIF, https://www.gbif.org/). Plant community traits can be calculated as the mean or median trait value of each species within a grid cell or ecoregion[17,18], but this approach does not consider species' relative abundances. Another common approach is to combine plant trait databases with plant community structure databases[19]. In practice, the trait of each species is represented by a single, globally averaged value[20,21], with the frequent missing species trait values of species imputed using genus or even family trait mean values or relying on other gap-filling techniques[22]. These approaches involve unknown levels of uncertainty, including that arising from statistical non-independence when mean trait values are used for species that inhabit more than one plant community[23]. Using locally measured traits and community structures would overcome these weaknesses[24].

Indeed, as numerous studies have demonstrated that stomatal traits show great intraspecific variation[25,26], using locally measured stomatal traits is essential for determining the stomatal morphology-environment relationships of plant communities. The mass ratio hypothesis predicts that ecosystem functioning should be largely determined by the plant traits of the dominant species within a community. As such, most studies focus on the community-weighted mean of plant traits[6]. Yet, according to the niche complementarity hypothesis predicting that resource niches may be used more completely when a community is functionally more diverse, functional diversity also plays an important role in ecosystem functioning[27,28]. For example, ecosystem multifunctionality can be strongly regulated by functional rarity and evenness of specific leaf area and plant height[29]. Therefore, in this study, we focused on four community-weighted trait moments—the mean, variance, skewness, and kurtosis—of stomatal morphology and their dependence on climate and soil factors (Supplementary Tables S1 and S2).

Here, we conduct a field survey in 57 natural ecosystems at a regional scale, which covers almost all vegetation types in the Northern Hemisphere (Fig. 1 and Supplementary Fig. S1). We establish a fine-resolution stomatal trait database, including stomatal density, size, and pore index of 4492 species-site combinations. Combined with community structure, we calculate community-weighted moments for stomatal traits. We hypothesize a trade-off between stomatal size and density across communities. Further, given the key roles of stomata in drought resistance and water use efficiency[3,30], we hypothesize that environmental variables related to water availability are the main drivers of stomatal traits. The environmental filtering hypothesis predicts that species with extreme trait values are more likely to be filtered out of a community with environmental stress, resulting in trait convergence in harsh conditions[31,32]. Thus, we further hypothesize that increasingly harsh and variable environments are associated with lower community stomatal trait diversity. We distinguish forests and grasslands, given their major divergence in life form composition and

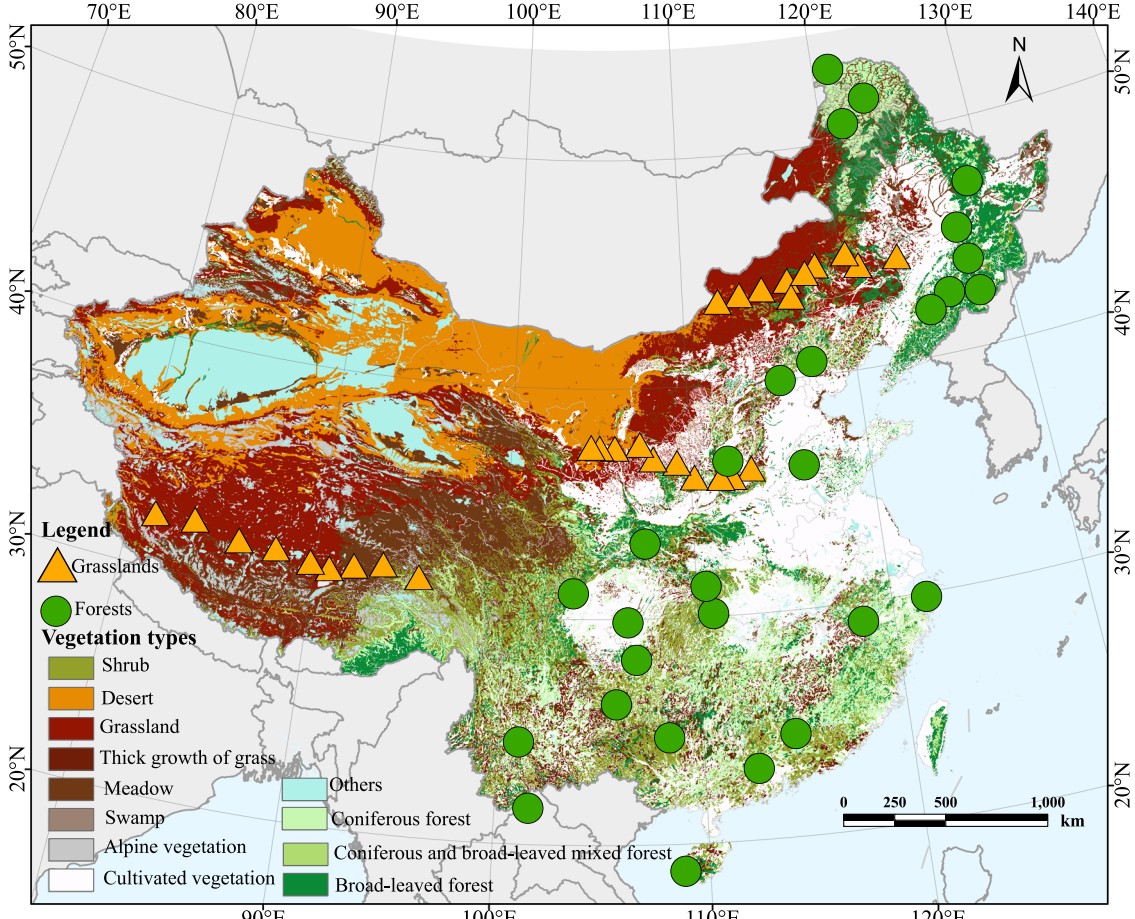

**Fig. 1 | Geographical distribution of the sampling sites in China.** The points on the map represent our sampling sites, and the vegetation types of our sampling sites are represented by different shapes and colors. The circle and triangle represent forest and grassland sampling sites, respectively. Source data are provided as a Source Data file.

ecosystem structure, and the greater environmental stress typically experienced by grasslands, such as severe drought, and, additionally, the higher water use efficiency of $C_4$ grasses. Given that one of the foundations of trait-based ecology is that trait-environment relationships tend to be consistent[33], we test whether stomatal trait-environment relationships are consistent in forests and grasslands. We hypothesize that grasslands tend to show greater convergence toward conservative stomatal traits, i.e., lower stomatal density and a smaller proportion of the leaf epidermis allocated to stomata, to reduce maximum rates of transpiration and improve water use efficiency.

Resolving the community-scale relationships between stomatal traits and environmental variables will provide key knowledge of community trait assembly and function under shifting climate.

## Results

### Overview of stomatal trait moments at community scale
Stomatal trait moments for 232 grassland plots and 108 forest plots were calculated (Supplementary Data 1). The community-weighted means of SD and SPI showed great variation across the plant communities, with 17-fold and 13-fold variation, respectively. The community-weighted means of SD and SPI for forests were significantly higher than for grasslands (Fig. 2 and Supplementary Table S3). The community-weighted mean of SL varied from 17.4 to 46.9 μm and did not differ on average between forests and grasslands (Fig. 2b).

The community-weighted variances of SD, SL, and SPI varied by over 10,000-fold, 1700-fold, and 7600-fold among the plant communities, respectively (Fig. 2d–f). Community-weighted mean and variances of stomatal traits were strongly positively correlated (Supplementary Fig. S2). Similar to the community-weighted mean values, community-weighted variances of SD and SPI were significantly higher for forests than grasslands, and the community-weighted variance of SL did not differ between forests and grasslands. By contrast, the community-weighted skewness and kurtosis of SD and SPI did not differ on average between forests and grasslands (Fig. 2g–i), but the community-weighted skewness and kurtosis of SL was higher for forests than grasslands.

We demonstrated a negative relationship between stomatal density and size at the community level (Fig. 3).

### Standardized effect sizes of stomatal trait moments
Overall, tests of standardized effect sizes (SESs) for stomatal moments indicated strong environmental filtering. Considering forests and grasslands together, the SESs for the variance in CWMs were positive, and those for the means of CWVs, CWSs, and CWKs were negative (Supplementary Table S4), and all of them were consistent with the predictions of the environmental filtering hypothesis. When considering only forests, the SESs for the variance in CWMs and for the means of CWSs and CWKs were consistent with the predictions of the environmental filtering hypothesis. When considering only grasslands, the SESs for the means of CWVs, CWSs, and CWKs were consistent with the predictions of the environmental filtering hypothesis.

### Relationships between stomatal trait moments and environmental variables
We found strong relationships between community-weighted means and variances of stomatal traits with environmental variables (Fig. 4, Supplementary Tables S5–16). Specifically, the community-weighted mean and variance of SD were mainly determined by precipitation in the warmest quarter, while those of SL were mainly determined by temperature seasonality. The community-weighted mean and variance of SPI were mainly determined by the mean temperature of the wettest quarter. Further, the community-weighted mean and variance of SD and SPI were also negatively correlated with soil sand content and

positively correlated with clay/silty content. The community-weighted skewness of SD, SL, and SPI was positively correlated with soil N content, the mean temperature of the warmest quarter, and isothermality, respectively, and the community-weighted kurtosis of SL and SPI were negatively correlated with soil pH and the mean temperature of the wettest quarter.

Overall, the community-weighted mean and variance of SD were mainly determined by precipitation factors, those of SL by temperature factors, and, for SPI, the community-weighted mean was mainly determined by temperature factors, and the community-weighted variance by both precipitation and temperature factors (Fig. 5). Climatic seasonality and climatic extreme variables played more important roles than climatic mean variables in driving stomatal trait moments.

Considering forests and grasslands separately revealed differences in their main drivers of stomatal trait moments (Supplementary Tables S5–S16). For both forests and grasslands, temperature seasonality was the main driver of community-weighted mean and variance of SL, but for other stomatal traits, the community-weighted moments of forests were mainly determined by temperature factors, whereas those of grasslands were more influenced by precipitation factors (Supplementary Figs. S3 and S4). Climatic seasonality and climatic extreme variables were important drivers of stomatal trait moments in both forests and grasslands.

## Discussion
Stomatal trait moments varied strongly both within and between forests and grasslands. The differences were consistent with theory and empirical findings at species scale, for which higher SD and SPI generally correspond to higher maximum photosynthetic rates and competitiveness, whereas lower SD and SPI are associated with reduced water loss in vegetation adapted to dry climates[4,34–36]. Grassland communities showed lower community-weighted means for SD and SPI. This may be an adaptation to conserve water given frequent dry periods in the hot summer growing season and shallower roots than forest trees. In particular, $C_4$ grass species tend to have lower SD and SPI[37]. Further, consistent with our hypothesis that harsh conditions would result in trait convergence[31,32], the community-weighted variances of SD and SPI were lower in grasslands than in forests. These findings indicate that the frequently droughted but competitive environments of grasslands tend to exclude species with lower and higher SD and SPI values, resulting in trait convergence. The community-weighted kurtosis of SL was lower in grasslands than that in forests. This result supports the expectation that stronger environmental stress would also lead to species being more evenly distributed within the community[38]. The community-weighted skewness of SL in forests was higher, consistent with a greater representation of functional rarity[29], associated with extreme trait values for non-dominant species or even rare species, as expected, given the environmental and functional heterogeneity of the forest community.

Across all plant communities, the observed variances in CWMs of the stomatal traits were higher than expected by chance (all SESs >0), and the observed means in CWVs, CWSs, and CWKs of these stomatal traits were lower than expected by chance (all SESs <0). These findings indicate that environmental filtering influenced community stomatal trait composition on a large scale. When forests and grasslands were analyzed separately, the means of CWSs and CWKs of both forests and grasslands were consistent with the predictions of the environmental filtering hypothesis. By comparison, the means of CWVs of forests and the variances of CWMs of grasslands did not reflect environmental filtering. A possible explanation for these patterns is that CWS and CWK were mainly influenced by rare phenotypes, which are likely to be filtered out of a community by environmental stress[32]. Therefore, compared with CWM and CWV, CWS and CWK showed more robust support for environmental filtering, especially in the narrower

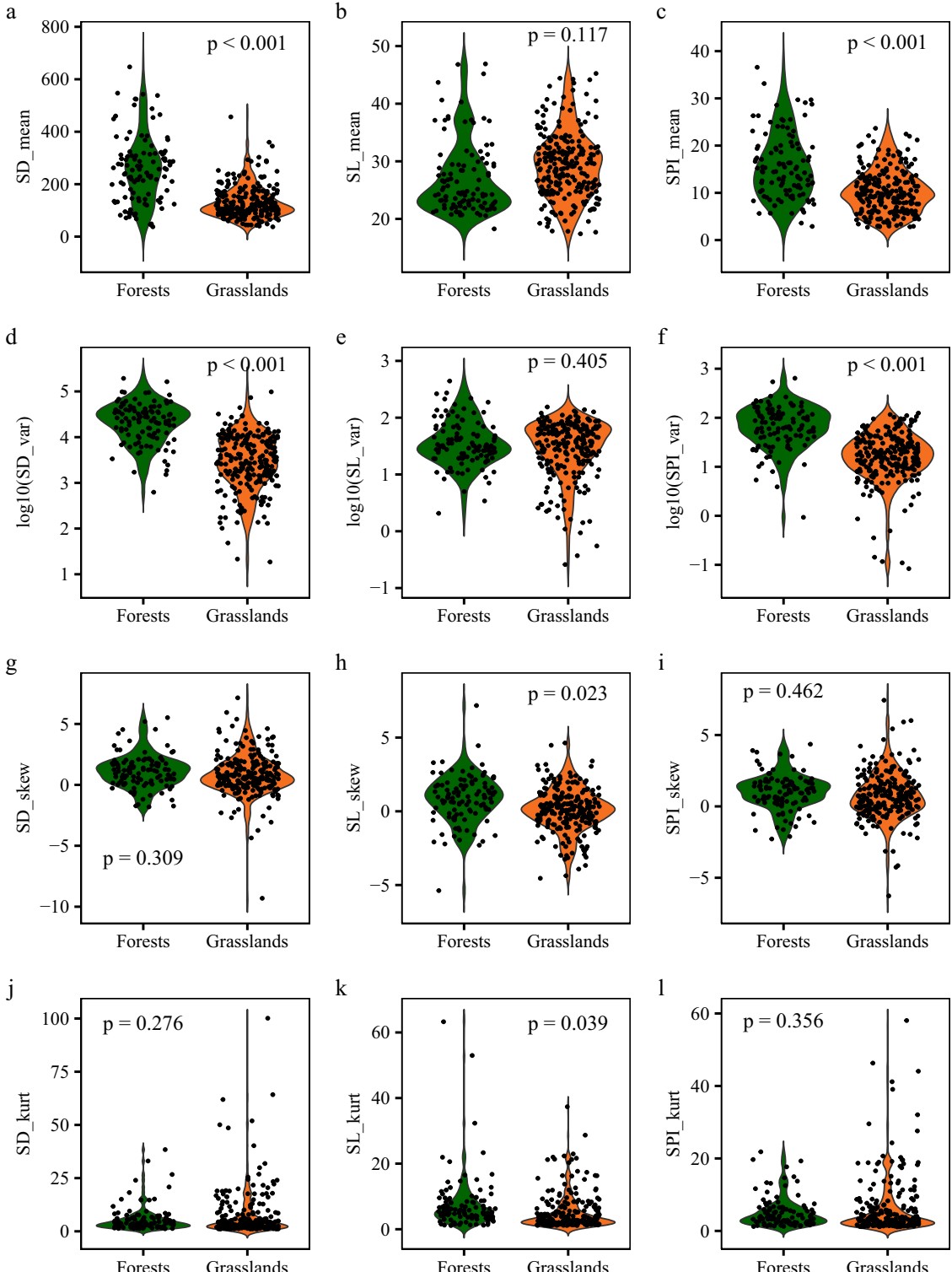

**Fig. 2 | Community-weighted trait moments of stomatal morphology in forests and grasslands.** Violin plots depict distributions of stomatal morphology moments using density curves, and the width of each curve corresponds with the approximate frequency of data points in each region. Violin plots of stomatal morphology moments for forests and grasslands are filled with different colors. Each point represents a forest or grassland plot. Statistical analysis was performed using linear mixed-effects models with vegetation types as a fixed factor and plot nested within sites as a random factor. For exact statistical values, see Supplementary Table S3. **a–l** Differences in community-weighted mean (**a–c**), variance (**d–f**), skewness (**g–i**), and kurtosis (**j–l**) of stomatal traits between forests and grasslands. SD_mean, community-weighted mean of stomatal density; SL_mean, community-weighted mean of stomatal length; SPI_mean, community-weighted mean of stomatal pore index. SD_var, community-weighted variance of stomatal density; SL_var, community-weighted variance of stomatal length; SPI_var, community-weighted variance of stomatal pore index. SD_skew, community-weighted skewness of stomatal density; SL_skew, community-weighted skewness of stomatal length; SPI_ skew, community-weighted skewness of stomatal pore index. SD_kurt, community-weighted kurtosis of stomatal density; SL_kurt, community-weighted kurtosis of stomatal length; SPI_kurt, community-weighted kurtosis of stomatal pore index. Source data are provided as a Source Data file.

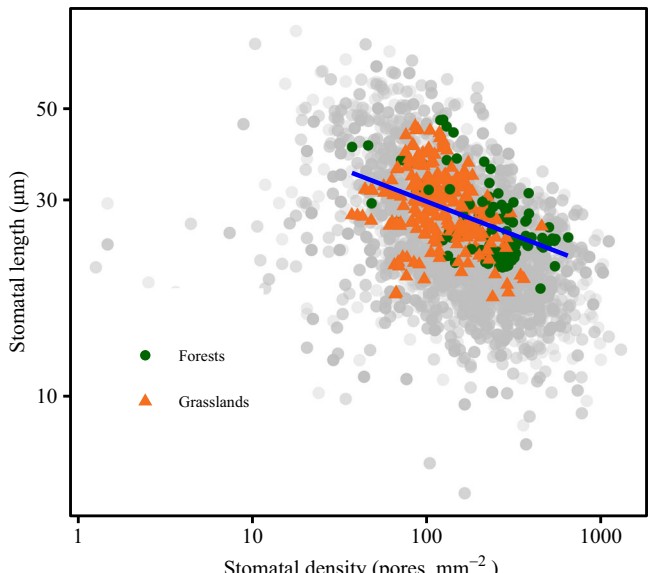

**Fig. 3 | Trade-off between stomatal density and size at the species (gray points) and community (color points) scales.** The blue line represents ordinary least square regression based on community-scale stomatal density and size (two-sided Pearson $r$-squared = 0.204, $p < 0.001$). Community-scale stomatal traits of forests and grasslands are represented by points with different shapes and colors. The circle and triangle represent forest and grassland sampling sites, respectively. Source data are provided as a Source Data file.

environmental gradients for each ecosystem type. This coincided with previous studies that used CWS and CWK to unveil ecological assembly rules[39].

We demonstrated at the across-community scale the trade-off between stomatal density and size that has been frequently described within and across species[5,40]. This trade-off has been explained based on the tendency across species to maximize gas exchange while reducing the epidermal space allocated to stomata[41]. Notably, variation in plant trait values enables co-occurring species to exploit different resources or the same resources at different spatial or temporal scales[42,43], contributing to the ability of species to co-exist in the same community. Similarity in trait-trait correlations at the species and community scales indicates that environmental and anatomical constraints on community assembly and species evolution are consistent[21]. This inference aligns with previous research investigating the scaling exponent of stomatal density versus size[44].

The community-weighted mean and variance of stomatal traits were strongly influenced by environmental variables. Consistent with our hypothesis that water availability was the main driver of stomatal traits, the community-weighted mean and variance of SD and SPI were strongly correlated with the precipitation of the warmest quarter. As the construction and maintenance costs of stomata are much higher than leaf epidermis cells[40,45,46], SD and SPI confer higher maximum photosynthetic rates[47]. Thus, high SD and SPI would confer high returns when water availability is high, explaining why the precipitation of the warmest quarter was the main environmental filter of SD and SPI. Given that soils with high sand and low clay/silty content generally have a low water retention capacity[48], plants growing in such soil would benefit from a low SD and SPI that would prevent excessive loss of water under dry conditions. Notably, the community-weighted mean and variance of SL were mainly associated with temperature seasonality and not with water availability. In this case, contrary to our hypothesis that harsh and variable environments would be associated with the lower stomatal trait diversity of plant communities, the community-weighted variance of SL was large under high temperature seasonality. This finding may reflect the dual functions of stomatal

size—with large SL contributing to maximum opening and gas exchange and small stomata to more rapid opening and closing of stomata. Overall, for each of the three stomatal traits (SD, SL, and SPI), the community-weighted means and variances were strongly positively correlated and mainly driven by the same climatic variables. This result indicates that directional and disruptive selection might combine to shape stomatal trait distribution within the community[49].

Compared with community-weighted mean and variance, the relationships between skewness and kurtosis of stomatal traits and environmental variables were much weaker. These weak associations of community-weighted skewness and kurtosis with macroclimate and coarse-scale soil properties are consistent with the abundance of extreme trait values associated with non-dominant species and rare species, mainly depending on the microenvironment[50]. This finding is consistent with a study of dryland that showed that the associations of plant height and specific leaf area with environmental variables decreased when explaining higher moments of trait distributions[20].

The community-scale relationships of stomatal traits with environment variables differed between forests and grasslands, in contrast to the frequent assumption of trait-based ecology that community-scale traits should show consistent relationships with environmental gradients[33,51]. Although the associations of community-weighted means and variances for SL and the environment were relatively consistent between vegetation types, those of SD and SPI were strongly dependent on vegetation type.

In contrast with a previous study of 17 plant traits at the global scale[21], in which the community-weighted mean and variance were poorly related to environmental variables, we found strong stomatal morphology-environment relationships of plant communities. Two reasons could explain this discrepancy. First, mismatching between plant traits and community structure in database studies might provide a weaker resolution of plant trait–environment relationships. Indeed, our study emphasizes that in situ sampling can reduce bias and provide stronger resolution. Second, the specific plant traits may vary in their patterning with the environment across communities, and stomatal traits may be more sensitive to environmental variation than other traits, such as size-related and economic traits. The strong associations of stomatal traits with the environment across scales from species to communities indicate their importance in providing evidence of both adaptation and community assembly in response to climate. Given our finding of strong climatic trends in stomatal traits and the demonstration in previous studies that stomatal traits strongly regulate ecosystem productivity[10], incorporating plant trait–environment relationships into dynamic global vegetation models could improve the predictions of climate change effects on ecosystem processes and functioning[52]. Indeed, as expected based on recent theoretical frameworks for the influence of traits on ecosystem processes at regional and continental scales[53,54], stomatal traits have been appreciated as important parameters for next-generation dynamic global vegetation models to predict ecosystem productivity under climate change.

We noted that a large part of the variance in stomatal trait moments was also accounted for by the random site factor (site, plots were nested within the site), indicating that similar climate and soil conditions support diverse plant communities with diverse stomatal trait assembly. Further, a large part of the variance in stomatal trait moments was not explained by environmental variables. Thus, future studies should consider local factors, such as microclimate, fine-scale soil properties, topography, and even site-specific biotic factors, to improve the predictions of stomatal trait moments at a continental scale.

In conclusion, we observed differences in stomatal trait moments between forests and grasslands, with lower community-weighted mean and variance of SD and SPI for grasslands than forests. We

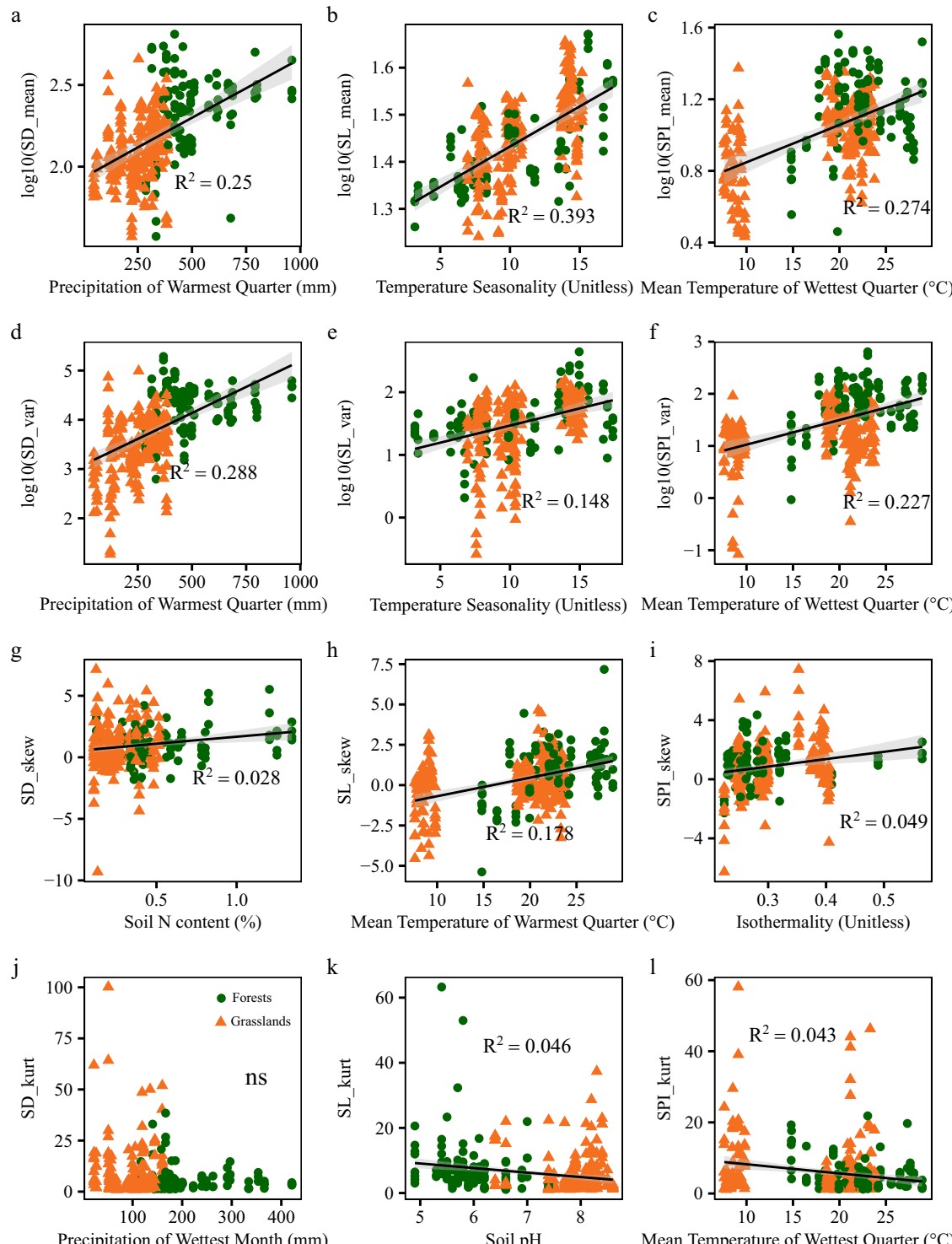

**Fig. 4 | Associations of community-weighted trait moments of stomatal morphology with environmental variables.** Stomatal trait moments of forests and grasslands are represented by different shapes and colors. The circle and triangle represent forest and grassland sampling sites, respectively. The solid lines represent the linear regressions, while the gray shading indicates the 95% confidence interval. All linear regressions are estimated using the linear mixed model with plot nested within sites as a random factor. $R^2$ is the marginal $R^2$ (fixed effects only). ns represents not significant. **a–l** Relationships between community-weighted mean (**a–c**), variance (**d–f**), skewness (**g–i**) and kurtosis (**j–l**) of stomatal traits and environmental variables. For exact statistical values, see Supplementary

Tables S5–S16. SD_mean, community-weighted mean of stomatal density; SL_mean, community-weighted mean of stomatal length; SPI_mean, community-weighted mean of stomatal pore index. SD_var, community-weighted variance of stomatal density; SL_var, community-weighted variance of stomatal length; SPI_var, community-weighted variance of stomatal pore index. SD_skew, community-weighted skewness of stomatal density; SL_skew, community-weighted skewness of stomatal length; SPI_skew, community-weighted skewness of stomatal pore index. SD_kurt, community-weighted kurtosis of stomatal density; SL_kurt, community-weighted kurtosis of stomatal length; SPI_kurt, community-weighted kurtosis of stomatal pore index. Source data are provided as a Source Data file.

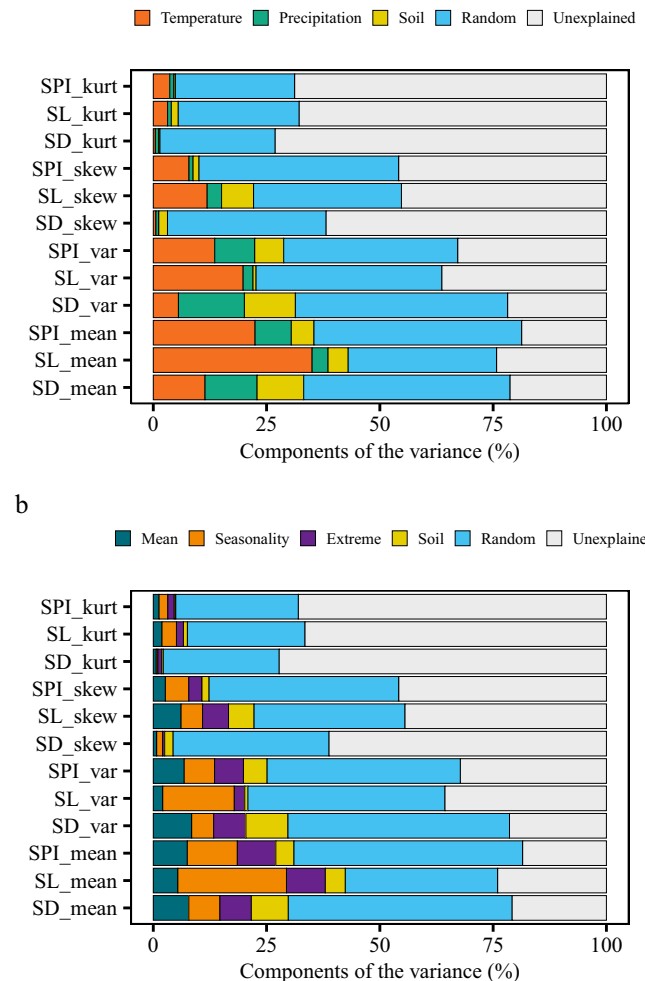

**Fig. 5 | Variance partitioning of stomatal trait moments.** Different colors represent different environmental groups, random factor, and unexplained variation. Statistical analysis was performed using linear mixed-effects models with environmental groups as fixed factors and plot nested within sites as random factors. **a** The relative importance of temperature factors, precipitation factors, soil factors, and random factors for stomatal trait moments (two-tailed statistical test). **b** The relative importance of climatic mean, climatic seasonality, climatic extreme, soil factors, and random factors for stomatal trait moments (two-tailed statistical test). SD_mean, community-weighted mean of stomatal density; SL_mean, community-weighted mean of stomatal length; SPI_mean, community-weighted mean of stomatal pore index. SD_var, community-weighted variance of stomatal density; SL_var, community-weighted variance of stomatal length; SPI_var, community-weighted variance of stomatal pore index. SD_skew, community-weighted skewness of stomatal density; SL_skew, community-weighted skewness of stomatal length; SPI_skew, community-weighted skewness of stomatal pore index. SD_kurt, community-weighted kurtosis of stomatal density; SL_kurt, community-weighted kurtosis of stomatal length; SPI_kurt, community-weighted kurtosis of stomatal pore index. Source data are provided as a Source Data file.

identified a trade-off between stomatal density and size both at the species and community scales, thus suggesting similar constraints by anatomy and environment across scales. Our results showed a strong influence on community-weighted mean and variance of stomatal traits by environmental variables, with weaker influences on community-weighted skewness and kurtosis. These results highlight the strong patterning of trait–environment relationships across communities at the continental scale. Given the use of stomatal traits to predict carbon and water fluxes and ecosystem productivity in vegetation models, the resolution of community-scale trends contributes

to the power to predict ecosystem productivity in response to future climate change.

## Methods

### Study sites
We selected a total of 57 typical natural ecosystems, including 29 grasslands and 28 forests. The study sites extend from 18.7 to 51.8 °N in latitude and from 81.2 to 128.9 °E in longitude and represent most of the vegetation types in the northern hemisphere, including cold-temperate coniferous forest, temperate coniferous and broad-leaved mixed forest, warm temperate deciduous broad-leaved forest, sub-tropical evergreen broad-leaved forest, tropical rain forest, meadow steppe, typical steppe, and desert steppe (Fig. 1). The mean annual temperature ranges from −6.6 to 22.4 °C, and mean annual precipitation (MAP) ranges from 146 to 1834 mm (Supplementary Fig. S1).

### Field sampling
The field survey was conducted in July and August from 2013 to 2018, the peak period of vegetation growth. Sampling plots were located within well-protected national nature reserves or national ecological observatory sites in areas of relatively continuous vegetation representative of the given site. Coordinates and altitude, plant species composition, and community structure were recorded for each plot. In forests, three or four experimental plots (30 × 40 m) were established. The height and diameter at breast height (DBH) of each individual tree were recorded. Specifically, the height of the small trees was measured using telescopic rods, and the height of the large trees was measured with the Blume-Leiss hypsometer or directly in permanent sample plots with a tower crane. Further, one or two shrub subplots (5 × 5 m) and two or four herb subplots (1 × 1 m) were nested in each experimental plot. Shrub height and basal diameter were measured. The organs and total biomass of individual trees and shrubs were calculated using species-specific allometric regressions based on the measured values of DBH or basal diameter and height. Species-specific allometric regressions were obtained from the books, "Carbon storage in forest ecosystems in China: biomass models"[55] and "Handbook of biomass models of common shrubs in China"[56]. When the allometric regression of specific species was not available, we used the allometric regression of the same genera or mixed-species equations of a forest. For herbs, above-ground parts of each species within the plots were harvested and the above-ground biomass was measured immediately after oven-drying[9]. In grasslands, eight plots (1 × 1 m) in each site were established, then the above-ground biomass of each species was measured using the harvest method[57]. Overall, 108 plots of 28 forests and 232 plots of 29 grasslands were investigated.

To collect ample replications of each species at a specific site, leaves were collected within and around the plots. For each species, 20–40 mature leaves were collected from at least four healthy individuals and mixed as a composite sample. All leaves of each species were placed in a sealed plastic bag and tagged with its name. All sample plastic bags were immediately stored in a cool box with ice. After the field sampling, eight to 10 leaves from the pooled sample of each species were cut into small pieces (1.0 × 0.5 cm) along the main vein and were fixed in FAA fixative (75% alcohol: formalin: glacial acetic acid: glycerin). Finally, 4492 species-site combinations were collected.

### Measurement of stomatal traits
Stomatal traits were measured using a scanning electron microscope (S-3400N, Hitachi, Japan). Most species in forests are hypostomatic[58], and given limitations by labor and expense, we only focused on the lower epidermis of forest species. Grasslands are open habitats and many species are amphistomatous, and thus, we focused on both epidermises of grassland species. For each species, three small pieces were selected from the pooled sample. Each replicate was photographed twice at different positions on the lower surface (forest

species) or both surfaces (grassland species); thus, we imaged each forest species six times and each grassland species six or 12 times. The leaf samples were photographed at 300–500 ×magnification depending on their stomatal morphology.

In each photograph, the number of stomata was recorded and five typical stomata were selected to measure the stomatal length (SL, μm) using an electronic image analysis software MIPS (Optical Instrument Co., Ltd., Chongqing, China). Stomatal density (SD, pores mm$^{-2}$) was calculated as the number of stomata per unit area. For amphistomatous species, SD was the sum of the upper and lower stomatal density, and SL was the average of the upper and lower stomatal length. The stomatal pore index (SPI, %) was used to represent the leaf surface covered by stomata[34], which was calculated as:

$$SPI = SD \cdot SL^2 \qquad (1)$$

## Stomatal traits at the community level

To scale up stomatal traits to the community scale, we used the total leaf biomass of each species in the plot to weigh species trait values and then calculated the distributions of stomatal traits, including community-weighted mean (CWM), variance (CWV), skewness (CWS), and kurtosis (CWK)[59]. CWM provides information on functional identity relating to the mass ratio hypothesis[6,60]. Whereas CWV, CWS, and CWK provide information on functional diversity relating to the niche complementarity hypothesis[20,27]. Stomatal trait moments were calculated as follows:

$$Trait_{mean} = \sum_1^n p_i Trait_i \qquad (2)$$

$$Trait_{var} = \sum_1^n p_i \left(Trait_i - _{Trait_{mean}}\right)^2 \qquad (3)$$

$$Trait_{skew} = \sum_1^n \frac{p_i \left(Trait_i - _{Trait_{mean}}\right)^3}{Trait_{var}^{\frac{3}{2}}} \qquad (4)$$

$$Trait_{kurt} = \sum_1^n \frac{p_i \left(Trait_i - _{Trait_{mean}}\right)^4}{Trait_{var}^2} \qquad (5)$$

where Trait_mean, Trait_var, Trait_skew, and Trait_kurt are CWM, CWV, CWS, and CWK of specific stomatal traits, respectively; $n$ is the number of species within a given plot, $p_i$ is the proportion of leaf biomass (forest plot) or above-ground biomass (grassland plot) of the $i$th plant species within a given plot, and $Trait_i$ represents SD, SL, or SPI of the $i$th plant species.

## Environmental variables

Environmental variables were derived from multiple global databases. We used climate data from the Climatic Research Unit at a high resolution (30 arc-seconds) for the 2011–2020 period[61] to calculate Bio1-Bio19, as defined by Fick and Hijmans[62]. We used the mean annual precipitation and potential evapotranspiration to calculate aridity index. We defined the growing season as being the set of consecutive months that satisfied the conditions: (1) monthly mean temperature ≥5°C, and (2) monthly precipitation/potential evapotranspiration ≥0.05[63], then growing-season temperature, precipitation, and aridity index were also included in our analysis. Key physicochemical properties (bulk density, soil total nitrogen, soil pH, and sand-silt-clay content) of topsoil (0–5 cm depth) were collected from the SoilGrids (global gridded soil information, https://soilgrids.org/). Further, soil

moisture was extracted from Meng et al.[64]. The 23 climatic variables and 7 soil variables used are provided in Supplementary Table S2.

## Statistical analyses

For each species at a given site, we first calculated the mean values of the stomatal traits. The stomatal traits were combined with community structure data to calculate CWM, CWV, CWS, and CWK of stomatal traits. The CWM and CWV of the stomatal traits were log-transformed for subsequent analyses. To investigate whether stomatal trait moments were associated with vegetation type (forests vs grasslands), we constructed a linear mixed-effects model using R package lme4[65], including site as a random effect and vegetation as a fixed effect: lmer (Trait moment ~ vegetation+(1|site)).

We tested for environmental filtering as the deviation of trait moments from random expectation. We randomized stomatal trait values across all species 500 times. For each run, we calculated stomatal trait moments with the randomized trait values but retained the species set and their abundances in each plant community intact[21]. For each stomatal trait, we calculated standardized effect sizes (SESs) for the variance in CWMs and for the mean in CWVs, CWSs, and CWKs. SES was calculated as

$$SES_{var(CWM)} = \frac{var(CWM_{obs}) - mean(var(CWM_{ran}))}{s.d.\left(var(CWM_{ran})\right)} \qquad (6)$$

$$SES_{mean(CWV, or CWS, or CWK)} =$$
$$\frac{mean(CWV_{obs}, or\ CWS_{obs}, or\ CWK_{obs}) - mean(CWV_{ran}, or\ CWS_{ran}, or\ CWK_{ran})}{s.d.\left(mean(CWV_{ran}, or\ CWS_{ran}, or\ CWK_{ran})\right)}$$
$$(7)$$

where var, mean, and s.d. represented computing the mean, variance, and standard deviation of the vector. Positive SESs for the variance of CWM and negative SESs for the means of CWV, CWS, and CWK would be consistent with the environmental filtering of stomatal traits[21]. These processes were also performed separately for forests and grassland.

We further investigated the potential correlations between the stomatal trait moments and 27 environmental variables and constructed a linear mixed-effects model as: lmer (Trait moment ~ environmental variables +(1|site)), and the marginal $R^2$ (fixed effects only) and conditional $R^2$ (both fixed and random effects) were calculated using R package MuMIn. Bivariate relationships with the highest marginal $R^2$ were graphed.

To further test how environmental variables drive community-weighted stomatal trait moments, 27 environmental variables used in this study were categorized as temperature, precipitation, and soil variables. As the aridity index was calculated using mean annual precipitation and was highly correlated with mean annual precipitation (two-sided Pearson $r = 0.96$, $p < 0.001$), it was grouped into precipitation variables. We then conducted PCA for each of the three environmental categories. The first two axes of the principal component analyses (PC1 and PC1) accounted for 96.5%, 94.1% and 71.6% of the variation in the temperature variables, precipitation variables, and soil variables, respectively. We then constructed a linear mixed-effects model as: lmer (Trait moment ~ T_PC1 + T_PC2 + P_PC1 + P_PC1+Soil_PC1+Soil_PC2 + (1|site)), where T, P, and Soil were temperature variables, precipitation variables, and soil variables, respectively. Using R package glmm.hp[66], we calculated the contributions of these predictors (including fixed and random effects), and then the resulting effects of each environmental predictor were grouped into their specific environmental categories.

Following Chen et al.[67], climatic variables were further divided into categories of climatic mean, climatic seasonality, and climatic extreme. Specifically, the "climatic mean" group included mean annual temperature, mean annual precipitation, and aridity index. The

"climatic seasonality" group included the mean diurnal range, isothermality, temperature seasonality, temperature annual range, and precipitation seasonality, and the "climatic extreme" group included the maximum temperature of warmest month, the minimum temperature of coldest month, the mean temperatures of the wettest, driest, warmest and coldest quarters, the precipitation of the wettest and driest months, and of the wettest, driest, warmest, and coldest quarters. We performed a PCA for each climatic variable group. The first two axes of the principal component (PC1 and PC1) accounted for 93.9%, 84.2% and 71.2% of the variation in the climatic mean, climatic seasonality, and climatic extreme groups, respectively. Therefore, to distinguish which climatic variable group was the strongest driver of the stomatal trait moments, we constructed a linear mixed-effects model as follows: lmer (Trait moment ~ mean_PC1+ mean_PC2+ seasonality_PC1+ seasonality_PC2+ extreme_PC1+ extreme _PC2 + + Soil_PC1+Soil_PC2+(1|*site*)). Using R package *glmm.hp*[66], we calculated the contributions of these predictors (including the fixed and random effect), and the results were grouped into their specific environmental groups.

The bivariate stomatal trait–environment relationships and variance partitioning in linear mixed-effects models were also performed separately for forests and grassland. All analysis codes used for the study are available in Supplementary Code 1.

### Reporting summary

Further information on research design is available in the Nature Portfolio Reporting Summary linked to this article.

## Data availability

The raw and processed stomatal trait data at the community level generated in this study are available in Supplementary Data 1 and at https://doi.org/10.6084/m9.figshare.23590332. Source data are provided with this paper.

## Code availability

The R codes used for analyses for each figure included in this paper can be accessed in Supplementary Code 1 and at https://doi.org/10.6084/m9.figshare.23590332.

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

## Acknowledgements

This work was supported by National Natural Science Foundation of China [31988102, to N.H.; 32201311, to C.L.], National Key R&D Program of China [2022YFF080210102, to N.H.], CAS Project for Young Scientists in Basic Research [YSBR-037, to N.H.], the national science and technology basic resources survey program of China [2019FY101300, to N.H.], and Key Laboratory of Ecology and Environment in Minority Areas (Minzu University of China), National Ethnic Affairs Commission [No. KLEEMA202302, to C.L.].

## Author contributions

N.H., G.Y. and C.L. planned and designed the research; C.L., Y.L., J.Z. and Q.Z. conducted fieldwork and collected data; C.L., Y.L., and L.S. analyzed data and wrote the manuscript; L.S., C.L. and K.Y. revised the manuscript.

## Competing interests

The authors declare no competing interests.
