## [Peer Review File · Nature Communications]

Reviewers' Comments:

Reviewer #1:

Remarks to the Author:

Dear Editor, Dear authors

The study proposed by Congcong Liu and colleagues deals with the question of the control of stomatal traits integrated at the community scale by climatic and edaphic variables. Based on companion studies (Liu et al 2022 PCE; Liu et al 2021 BR; He et al 2022 TPS), the authors argued that studying traits at community scale is required to better predict the functioning at ecosystem scales and as such they focused on the two first moments of trait distribution. The authors studied stomatal density, stomatal size and their products of ca 80 species within 340 communities and across 57 sites. They tested if trait community means and variances covaried with the annual mean, seasonality and extremes of site temperature and precipitation as well as the mean of edaphic property. They postulated that stomatal traits should all covary with precipitation, whereas they should not covary with temperature. Finally, they postulated that trait variance should be rather determined by climate extremes and seasonality than by climate annual means. They showed that mean and variance of stomatal density did covary with precipitation, whereas the two moments of stomatal size covaried with temperature. As the stomatal density usually trade-offs with the stomatal size across species, showing that these two traits followed different climatic determinisms across communities is an important result of their study. In addition, they highlighted that community trait moments were more related with the seasonality and extreme of climatic variables rather than to their annual means. Whereas results are interesting, I found that the paper suffers from three major issues, which prevent publication. Let me explain them:

- First, the originality of the study in comparison with the series of recent companion papers by the same team is not straightforward. It seems that part of their database overlaps with the one used in the companion papers. The authors should explained the originality of this new manuscript. In Liu et al 2022 PCE, the authors argued that the four moments of trait community distribution are important, whereas here kurtosis and skewness are totally ignored with no explanation in the present study. In Liu et al 2021 BR, they showed that stomatal density and stomatal size covaried negatively. In the present study, they presumed the same quoting their previous paper but without clearly showing it. Overall, I recommend to better justifying and delineating the present study in regards with the previous ones.

- Second, I found many places where the rationale/justification is difficult to follow and rather weak:

- o L61-62: '...the study contained limited economic traits and size-related traits in forests.

Therefore, we further explore other trait-environment relationships...'. To my opinion this seems to be a poor justification to study stomatal trait. I would recommend a better/stronger justification on the interest in community stomatal traits.

- o L72-75: 'Previous study have identified plant traits related to drought tolerance are strongly correlated with precipitation 24, 25, 26. Therefore, stomatal traits at the community level should be mainly determined by precipitation rather than temperature.' The reference 24 concerns leaf turgor loss point, the reference 25 does not compare temperature vs precipitation and the reference 26 only concerns one species. To my opinion, this central assumption is not well supported by these references and overall not well justified. Based on Raven 2002, one could rather assume that stomatal traits can be similarly constrained by the two climate variables temperature and precipitation that modulate the climate aridity. For instance, Wright et al 2017 Science showed that leaf size is both determined by temperature and precipitation which interact to avoid leaf over-heating during the day. I recommend a better justification of this central assumption.

- o L81-82: 'We aim to explore stomatal trait-environment relationships of plant communities in forests and grasslands...'. Please justify why focusing on these two vegetation types and why they need to be analysed separately.

- o L130-132: 'Most species in forests are amphistomatous and limited by labor and expense. Therefore, we only focus on the lower epidermis of forest species. Grasslands are open habitats and many species are amphibious. Therefore, we focused on the upper and lower epidermis of grassland species.' If species in forest are amphistomatous, it means that they have stomata on both sides of the leaf, then it is unclear why focusing on one side only. In addition, it is not clear either why amphibious species should be studied on both sides. This is crucial to better explain this

part as it may change the results a lot.

o L147-152: 'Functional identity and diversity are collectively known as functional composition, and are calculated as the community-weighted mean (CWM) and variance (CWV), respectively. CWM is...'. Functional diversity does not concern variance only. Recent studies including LeBagoussePinguet et al 2021 PNAS as well as another study from the same author teams, recognised that the third and fourth moments are required to be studied when focusing on functional diversity, especially when the distribution does not follow a normal law. I would strongly recommend showing the trait community distribution and better justifying their choice on selecting the two first moments only.

o L161ff: Environmental variables were directly extracted from global database, Worldclim for climate and Global Soil Database for soil. There was no justification on these choices, whereas other climate and soil databases exist (CRU for climate, soilgrids for soils). In addition the choices of variables is not well justified in regards with leaf traits. For instance, trait-environment relationships are most of the time studied under the vegetation growing condition period rather than under annual mean. This is particularly relevant for this dataset as it includes boreal and tundra sites where the annual mean would be particularly not relevant for studying stomatal traits. Finally, the choice of soil variables seems unrelated with the known functioning of stomatal traits. For instance, bulk density or soil available water content could be more justified than soil total N or soil pH when focusing on stomatal traits.

- To my opinion, the work does not support the conclusions and claims. Authors claimed that stomatal size covaried negatively with stomatal density at the community level. Except the PCA that may had been shaped artificially by the mathematical product of stomatal size and density (see below), i.e. the stomatal pore index, there is no demonstration on this claim in the present manuscript. This point is mandatory as it is the core of the interpretations and the main message in the discussion section. I strongly recommend to better highlighting this trade-off at both species and community scales to then building on it.

Considering these three major issues and the requirements of novelty and scientific impact by Nature Communications, I recommend then to clarify these issues before any publication.

Further comments:

- L99-100 and 104-106 are redundant. Please remove the information at one place.

- L173ff:

o PCA of traits: As stomatal pore index was the product of stomata length and density, I think that the result of the PCA was strongly and artificially shaped by mathematical constrains rather than by the natural variation of SL and SD. I would recommend to remove SPI from the PCA.

o It is not clear how the authors dealt with the inflation factor in the different models as many climatic variables should covary. Please give information on this aspect.

o There were several plots in each site. This should have been included in the random factor nested within site. Is there a particular reason why authors did not do that?

o There are many results presented in this paragraph. This should go into the result section.

- L319ff: This paragraph is particularly difficult to follow. We don't know much about the stress-dominance hypothesis and its relationship with stomata. Seasonality is not explained and then it is difficult to understand the relationship with stomatal traits. Please give more interpretation on the different relationships discussed.

- Figure 3-4: Please use different shapes for distinguishing grasslands and forests, as they appears the same on white-black printed document.

- Figure 5: There was a huge variation (30-40%) explained by the random factor site. Authors should discuss what could be related with such variation across sites.

Regards

Vincent Maire

Reviewer #2:

Remarks to the Author:

Paper by Liu et al., titled "Stomatal morphology-environment relationships of plant communities"

is original and novel study greatly fitting for the scope of Nature Communications. It explores community level variability of stomatal morphology (stomatal size and density, stomatal pore index) across multiple forest and grassland ecosystems in China. Authors also explored impact of environmental variables (mean, extreme, seasonality) on variability of stomatal morphology. The scale of the field measurements and plant material sampling is quite impressive and the presented data represent a significant contribution to the scientific community. The results are very interesting and improve our knowledge of environmental drivers of stomatal morphological traits. The paper is overall well written, but English proofreading could help with the stylistic quality of the text and fix minor grammatical errors. Abstract is well composed and highlights the main findings of the paper. Keywords capture the contents of the paper very well. Introduction is quite extensive and provides great overview of problematics for the reader. The materials and methods chapter is well written, with detailed description of study sites, sampling approach, measuring method of stomatal morphology and statistical analyses. The visualizations of results are clear but quality of the figures must be improved. The supplementary materials are also quite extensive, providing further details to reader if needed. Discussion is adequate, but authors could expand on the comparison between forests and grasslands, as well as, include more recent papers focused on the environmental impacts on stomatal morphology. I suggest minor revision of the paper and acceptance in Nature communications, after the authors address my comments:

Introduction:

I suggest authors to expand the introduction with focus on significance of stomatal morphology for characterization of the communities. Stomatal morphology (both density and size) affect the drought resistance and WUE of the plants. This highlights the practical importance of this study!

Liu, C., et al., 2018. Variation of stomatal traits from cold temperate to tropical forests and association with water use efficiency. <https://doi.org/10.1111/1365-2435.12973>

Amitrano, C., et al. 2021. Reducing the Evaporative Demand Improves Photosynthesis and Water Use Efficiency of Indoor Cultivated Lettuce. <https://doi.org/10.3390/agronomy11071396>

Stojnić, S., et al. 2019. The use of physiological, biochemical and morpho-anatomical traits in tree breeding for improved water-use efficiency of *Quercus robur* L. <https://doi.org/10.5424/fs/2019283-15233>

Pitaloka, M.K., et al., 2022. Induced Genetic Variations in Stomatal Density and Size of Rice Strongly Affects Water Use Efficiency and Responses to Drought Stresses.

<https://doi.org/10.3389/fpls.2022.801706>

Materials and methods:

Line 114: Above-ground parts were harvested, not was harvest.

Line 116: Similar to the herb subplots... Or other phrasing.

Line 125: There is no shrinkage of the fixed samples in the alcohol solution which could affect the size and density of the stomata?

Line:130: Species are limited by labor and expense? Please reformulate the sentences.

Line 145: Please include the formula for the SPI calculation. Is it total sample area divided by $SD \cdot LA^2$?

Results:

Why did you choose to analyse the data only for forest and grassland categories and not for all vegetation types as you are mentioning at line 96: "cold-temperate coniferous forest, temperate deciduous forest, subtropical evergreen forest, tropical rain forest, meadow steppe, typical steppe, and desert steppe"? Ultimately its authors decision how they conceptualize their paper, I am just curious why you chose the simplified design of the data analysis.

Figure 2: The weighted averages for SD ranged from 37 to 456 pores per mm (a), but the weighted variance (d) is ranging up to 100 000 pores per mm ($\log_{10}(100000) = 5$)? Is that correct? Same for the other two parameters.

Discussion:

I am missing comparison between the forests and grasslands. Why do the forest communities have larger stomatal density than the grassland communities? Is this driven only by the climate, or also by the prevailing species with different life form (trees vs grasses)?

I suggest authors to expand their discussion and include more recent studies focused on environmental impact of climate on stomatal morphology, e.g.:

Petrík, P., et al. 2020. Stomatal and Leaf Morphology Response of European Beech (*Fagus sylvatica* L.) Provenances Transferred to Contrasting Climatic Conditions.

<https://doi.org/10.3390/f11121359>

Lin, Y., et al. 2021. Leaf traits from stomata to morphology are associated with climatic and edaphic variables for dominant tropical forest evergreen oaks.

<https://doi.org/10.1093/jpe/rtab060>

Sun, J., et al. 2021. Spatial variation of stomatal morphological traits in grassland plants of the Loess Plateau. <https://doi.org/10.1016/j.ecolind.2021.107857>

Petrík, P., et al. 2022. Interannual adjustments in stomatal and leaf morphological traits of European beech (*Fagus sylvatica* L.) demonstrate its climate change acclimation potential.

<https://doi.org/10.1111/plb.13401>

Line 337: You achieved 32% explanatory power for SD (best climatic parameter). Is it strong enough to be a contradictory result to Bruelheide et al. 2018? Your findings about environmental drivers of stomatal morphology are great, but still the source for major part of variance is unknown/random as shown in your figure 5.

Conclusion:

If you will expand your discussion with comparison between forests and grasslands, it would be fitting to mention that grasslands, which are better adapted to drier conditions have lower stomatal density than forest species.

Reviewer #1 (Vincent Maire)

Dear Prof. Maire,

We greatly appreciated your comments and advice, all of which we have addressed, further improving the rigor of our manuscript.

The study proposed by Congcong Liu and colleagues deals with the question of the control of stomatal traits integrated at the community scale by climatic and edaphic variables. Based on companion studies (Liu et al 2022 PCE; Liu et al 2021 BR; He et al 2022 TPS), the authors argued that studying traits at community scale is required to better predict the functioning at ecosystem scales and as such they focused on the two first moments of trait distribution. The authors studied stomatal density, stomatal size and their products of ca 80 species within 340 communities and across 57 sites. They tested if trait community means and variances covaried with the annual mean, seasonality and extremes of site temperature and precipitation as well as the mean of edaphic property. They postulated that stomatal traits should all covary with precipitation, whereas they should not covary with temperature. Finally, they postulated that trait variance should be rather determined by climate extremes and seasonality than by climate annual means. They showed that mean and variance of stomatal density did covary with precipitation, whereas the two moments of stomatal size covaried with temperature. As the stomatal density usually trade-offs with the stomatal size across species, showing that these two traits followed different climatic determinisms across communities is an important result of their study. In addition, they highlighted that community trait moments were more related with the seasonality and extreme of climatic variables rather than to their annual means.

Response: We are very grateful for this thorough understanding and synthesis of our work.

Whereas results are interesting, I found that the paper suffers from three major issues, which prevent publication. Let me explain them:

- First, the originality of the study in comparison with the series of recent companion papers by the same team is not straightforward. It seems that part of their database overlaps with the one used in the companion papers. The authors should explain the originality of this new manuscript. In Liu et al 2022 PCE, the authors argued that the four moments of trait community distribution are important, whereas here kurtosis and skewness are totally ignored with no explanation in the present study. In Liu et al 2021 BR, they showed that stomatal density and stomatal size covaried negatively. In the present study, they presumed the same quoting their previous paper but without clearly showing it. Overall, I recommend to better justifying and delineating the present study in regards with the previous ones.

Response: We are grateful to the reviewer for the thorough reading of our previous work. As the reviewer described, we have previously published papers on stomatal distributions in the Journal of Experimental Botany (“Contrasting adaptation and optimization of stomatal traits across communities at continental scale”) and BR (“Scaling between stomatal size and density in forest plants”). This new paper contains a large amount of novel data, and the main and featured findings too are novel, unpublished elsewhere. The species sample in JXB (from 9 forest sites, dataset

completed in 2014) is indeed included as part of the species sample in this article (from 29 grassland sites and 28 forest sites, dataset completed in 2022). In the JXB paper, we mainly focused on the **maximum stomatal conductance (g), stomatal area fraction (f), and their ratio (i.e., the space-use efficiency, e)**, across nine forests. In this paper, we focus on different variables, i.e., the **stomatal density, stomatal size**, and their product-stomatal pore index.

We agree that the four moments of trait community distribution are important and have now added the community-weighted kurtosis and skewness to this paper.

With respect to the trade-off between stomatal density and size across diverse plant species, it is true we did not focus on this in the present paper. We have considered that trend to be well-established; it was first published in 1865, and has been the focus of many papers since. In the BR paper, we addressed a long-standing, but unsolved question about the evolutionary mechanism for this trade-off with novel theory and data. That BR paper included stomatal traits of 28 forest sites and additionally, stomatal traits of forest species that we compiled from peer-reviewed papers, and focused on that one specific question (the evolution of the stomatal size vs density trade-off). This current paper contains the additional data from 29 grassland sites, and additional novel data for the 29 grassland sites and 28 forest sites, including community composition, and scaled-up community metrics for stomatal density and size. Thus, the dataset in this paper is by majority highly novel, and its focus entirely distinctive. In particular, whereas the BR paper focused on evolutionary relationships between traits at the species level, we here focus on trait-trait correlations at the community level, which are the results of community assembly.

Collecting these stomatal traits datasets have encompassed nearly 10 years, resulting in an unprecedented volume and significance of data. Overall, this study is the first to reveal stomatal trait–environment relationships of plant communities across broad geographic gradients, importantly transforming knowledge of trait–environment relationships at this large spatial scale, which is gaining increasing interest and importance for global ecology.

We are grateful to the reviewer for the chance to explain all these important aspects and can assure them that we are here publishing an enormous quantity of novel data and calculated metrics and entirely novel findings, not published elsewhere.

- Second, I found many places where the rationale/justification is difficult to follow and rather weak:

- o L61-62: ‘...the study contained limited economic traits and size-related traits in forests. Therefore, we further explore other trait-environment relationships...’. To my opinion this seems to be a poor justification to study stomatal trait. I would recommend a better/stronger justification on the interest in community stomatal traits.

Response: We agree strongly with this comment, and have rewritten the rationale:

Studies using locally measured traits and community structures would overcome these weaknesses
23.

As numerous studies have demonstrated that the stomatal traits showed great intraspecific variation
24, 25, *determining the stomatal morphology-environment relationships of plant communities using locally measured stomatal traits is essential.*

- o L72-75: ‘Previous study have identified plant traits related to drought tolerance are strongly

correlated with precipitation 24, 25, 26. Therefore, stomatal traits at the community level should be mainly determined by precipitation rather than temperature.’ The reference 24 concerns leaf turgor loss point, the reference 25 does not compare temperature vs precipitation and the reference 26 only concerns one species. To my opinion, this central assumption is not well supported by these references and overall not well justified. Based on Raven 2002, one could rather assume that stomatal traits can be similarly constrained by the two climate variables temperature and precipitation that modulate the climate aridity. For instance, Wright et al 2017 Science showed that leaf size is both determined by temperature and precipitation which interact to avoid leaf overheating during the day. I recommend a better justification of this central assumption.

Response: We were grateful for this point, and agree. We have revised the hypothesis to address this point and modified the text:

Given that the roles of stomata in drought resistance and water use efficiency 3, 29, we hypothesized that environmental variables related to water availability were the main drivers of stomatal traits. The environmental filtering hypothesis predicts that species with extreme trait values are more likely to be filtered out of a community with environmental stress, resulting in trait convergence in harsh conditions 30, 31. Thus, we further hypothesized that increasingly harsh and variable environments would be associated with lower community stomatal trait diversity. Resolving the community scale relationships between stomatal traits and environmental variables will provide key knowledge of community trait assembly and function under shifting climate.

o L81-82: ‘We aim to explore stomatal trait-environment relationships of plant communities in forests and grasslands...’. Please justify why focusing on these two vegetation types and why they need to be analysed separately.

Response: We have rewritten it as follows, and here we did not emphasize forests and grasslands *To accomplish this goal, we conducted a field survey in 57 natural ecosystems at a regional scale, which covered almost all vegetation types in the Northern Hemisphere. Then, an unprecedented fine-resolution stomatal trait database was established, including stomatal density, size, and pore index of 4492 species-site combinations. Combined with community structure, community-weighted moments for stomatal traits were calculated.*

In this study, we contrasted forest and non-forest (grassland) systems, given their major divergence in life form composition and ecosystem structure

o L130-132: ‘Most species in forests are amphistomatous and limited by labor and expense. Therefore, we only focus on the lower epidermis of forest species. Grasslands are open habitats and many species are amphibious. Therefore, we focused on the upper and lower epidermis of grassland species.’ If species in forest are amphistomatous, it means that they have stomata on both sides of the leaf, then it is unclear why focusing on one side only. In addition, it is not clear either why amphibious species should be studied on both sides. This is crucial to better explain this part as it may change the results a lot.

Response: We thank the reviewer for finding this typo—most species in forests are hypostomatic (i.e., stomata only on the lower side). We have corrected this error.

o L147-152: ‘Functional identity and diversity are collectively known as functional composition, and are calculated as the community-weighted mean (CWM) and variance (CWV), respectively.

CWM is...'. Functional diversity does not concern variance only. Recent studies including LeBagoussePinguet et al 2021 PNAS as well as another study from the same author teams, recognised that the third and fourth moments are required to be studied when focusing on functional diversity, especially when the distribution does not follow a normal law. I would strongly recommend showing the trait community distribution and better justifying their choice on selecting the two first moments only.

Response: We agree with the reviewer, and in this revised version, we have focused not only on the community-weighted mean and variance, but also kurtosis and skewness. We explain our approach in the following revised lines:

Line73-81 Influenced by the mass ratio hypothesis, which predicts that ecosystem functioning should be largely determined by the plant traits of the dominant species within a community, most studies focus on the community-weighted mean of plant traits 5. According to the niche complementarity hypothesis, which predicts resource niches may be used more completely when a community is functionally more diverse, functional diversity also plays an important role in ecosystem functioning 26, 27; for example, ecosystem multifunctionality can be strongly regulated by functional rarity and evenness of specific leaf area and plant height 28. Therefore, in this study, we focused on four community-weighted trait moments (including mean, variance, skewness, and kurtosis) of stomatal morphology, and their dependence on climate and soil factors.

Line 152-155 To scale up stomatal traits to the community scale, we used the total leaf biomass of each species in the plot to weight species trait values, and then calculated the distributions of stomatal traits, including community-weighted mean (CWM), variance (CWV), skewness (CWS), and kurtosis (CWK)

All the figures, tables, and statistics were modified in this version to include these additional metrics.

o L161ff: Environmental variables were directly extracted from global database, Worldclim for climate and Global Soil Database for soil. There was no justification on these choices, whereas other climate and soil databases exist (CRU for climate, soilgrids for soils). In addition the choices of variables is not well justified in regards with leaf traits. For instance, trait-environment relationships are most of the time studied under the vegetation growing condition period rather than under annual mean. This is particularly relevant for this dataset as it includes boreal and tundra sites where the annual mean would be particularly not relevant for studying stomatal traits. Finally, the choice of soil variables seems unrelated with the known functioning of stomatal traits. For instance, bulk density or soil available water content could be more justified than soil total N or soil pH when focusing on stomatal traits.

Response: We accepted your advice, and now use the suggested databases, i.e., CRU for climate variables and soilgrids for soil variables (soilgrids). With these new data, we have re-analyzed all trends, and remade all the figures and tables. The main findings have remained substantially identical. Indeed, this makes sense because the climate variables are strongly correlated across the two databases. WorldClim included 19 Bioclimatic variables (BIO1-BIO19) for 1970-2000; when

we used the CRU monthly temperature and precipitation (2011-2020) to calculate the BIO1-BIO19, we found the following correlations:

Climate	r	Climate	r
BIO1	0.9902955	BIO11	0.9923353
BIO2	0.9500666	BIO12	0.9851045
BIO3	0.9860584	BIO13	0.9420328
BIO4	0.9937014	BIO14	0.972097
BIO5	0.9796619	BIO15	0.8458449
BIO6	0.9881573	BIO16	0.9682974
BIO7	0.982813	BIO17	0.9693506
BIO8	0.9829816	BIO18	0.9498292
BIO9	0.9722548	BIO19	0.9786687
BIO10	0.988686	AI	0.9781156

We agreed with reviewer that CRU climate variables are much better, and included the years of our field period (2013-2018).

Besides, we also agreed that we should consider climate variables of growing seasons

We defined the growing season as being the set of consecutive months that satisfied the conditions: (a) monthly mean temperature $\geq 5^{\circ}\text{C}$, and (b) monthly precipitation/potential evapotranspiration ≥ 0.05 (Wright et al., 2017).

We also included the bulk density and soil moisture in this new version.

Some key details were added in the revision:

*Environmental variables were derived from multiple global databases. We used climate data from the **Climatic Research Unit** at a high-resolution (30 arc-seconds) for the 2011 - 2020 period to calculate Bio1-Bio19, as defined by Fick and Hijmans 38. We used the mean annual precipitation and potential evapotranspiration to calculate aridity index. We defined **the growing season** as being the set of consecutive months that satisfied the conditions: (a) monthly mean temperature $\geq 5^{\circ}\text{C}$, and (b) monthly precipitation/potential evapotranspiration ≥ 0.05 (Wright et al., 2017), then growing-season temperature, precipitation, and aridity index were also included in our analysis. Key physico-chemical properties of topsoil (bulk density, soil total nitrogen, soil pH, and sand-silt-clay content) were collected from the SoilGrids (global gridded soil information, <https://soilgrids.org/>). Further, **soil moisture** was extracted from Meng, Mao 39. The 23 climatic variables and 7 soil variables used are provided in Table S1.*

- To my opinion, the work does not support the conclusions and claims. Authors claimed that stomatal size covaried negatively with stomatal density at the community level. Except the PCA that may had been shaped artificially by the mathematical product of stomatal size and density (see below), i.e. the stomatal pore index, there is no demonstration on this claim in the present manuscript. This point is mandatory as it is the core of the interpretations and the main message in the discussion section. I strongly recommend to better highlighting this trade-off at both species and community scales to then building on it.

Response: We agree and have added a new figure to show the trade-off between stomatal density and size at the species and community levels.

Further comments:

- L99-100 and 104-106 are redundant. Please remove the information at one place.

Response: Thank you. We have deleted the former sentence in this new version.

- L173ff:

o PCA of traits: As stomatal pore index was the product of stomata length and density, I think that the result of the PCA was strongly and artificially shaped by mathematical constrains rather than by the natural variation of SL and SD. I would recommend to remove SPI from the PCA.

Response: We removed the SPI from the PCA as recommended.

o It is not clear how the authors dealt with the inflation factor in the different models as many climatic variables should covary. Please give information on this aspect.

Response: We appreciated this request. In multiple linear regressions, we used `glmm.hp` R package to calculate the contribution of each independent variable to the dependent variable. Here is the description of the `glmm.hp` R package:

*Generalized linear mixed models (GLMMs) have been widely used in contemporary ecology studies. However, **determination of the relative importance of collinear predictors** (i.e. fixed effects) to response variables is one of the challenges in GLMMs. Here we developed a novel R package, `glmm.hp`, to decompose marginal R² explained by fixed effects in GLMMs. The algorithm of `glmm.hp` is based on the recently proposed approach “**average shared variance**” that is used for multivariate analysis. We explained the principle and demonstrated the use of this package by simulated dataset. The output of `glmm.hp` shows individual marginal R²s that can be used to evaluate the relative importance of predictors, which sums up to the overall marginal R².*

In other words, `glmm.hp` R package enables the evaluation of collinearity of factors.

We also checked the VIF of our multiple linear regressions using `vif` function in `car` R package.

For the model

```
lmer(Trait~temperature1+temperature2+precipitation1+precipitation2+soil1+soil2+(1|Site),
data=data)
```

The result was

temperature1	temperature2	precipitation1	precipitation2	soil1	soil2
2.663458	1.483762	4.349655	1.649147	2.680881	3.225530

```
lmer(Trait~climatic_mean1+climatic_mean2+climatic_seasonality1+climatic_seasonality2+climatic_extreme1+climatic_extreme2+soil1+soil2+(1|Site), data=data2)
```

The result was

climatic_mean1	climatic_mean2	climatic_seasonality1	climatic_seasonality2
85.108519	29.322994	4.856310	5.765084
climatic_extreme1	climatic_extreme2	soil1	soil2
92.329002	28.351529	3.473808	2.212864

As we can see, in the second multiple linear regressions, there is strong collinearity between the independent variables. If you strongly disagree with `glmm.hp` to deal with this, we could reanalyze the second multiple linear regressions using other methods in the next version.

o There were several plots in each site. This should have been included in the random factor nested within site. Is there a particular reason why authors did not do that?

Response: We did do this, and have more explicitly described how in each analysis, “site” was included as a random factor.

o There are many results presented in this paragraph. This should go into the result section.

Response: We followed your advice, and moved some sentences into the Results section.

- L319ff: This paragraph is particularly difficult to follow. We don't know much about the stress-

dominance hypothesis and its relationship with stomata. Seasonality is not explained and then it is difficult to understand the relationship with stomatal traits. Please give more interpretation on the different relationships discussed.

Response: Here, it was modified as follows:

In this case, contrary to our hypothesis that harsh and variable environments would be associated with lower the stomatal trait diversity of plant communities, the community-weighted variance of SL was large under high temperature seasonality. This finding may reflect the dual functions of stomatal size—with large SL contributing to maximum opening and gas exchange, and small stomata to more rapid opening and closing of stomata. Overall, for each of the three stomatal traits (SD, SL, and SPI), the community-weighted means and variances were strongly positively correlated, and mainly driven by the same climatic variables, indicating that directional and disruptive selection might combine to shape stomatal trait distribution within the community 58.

- Figure 3-4: Please use different shapes for distinguishing grasslands and forests, as they appears the same on white-black printed document.

Response: We took this advice, and used different shapes to distinguish vegetation types, circles for forests, and triangles for grasslands. For example, Figure 3 was modified as follows:

- Figure 5: There was a huge variation (30-40%) explained by the random factor site. Authors should discuss what could be related with such variation across sites.

Response: We added a paragraph to explain this point:

We note that a large part of the variance in stomatal trait moments was also accounted for by the random site factor (site, plots were nested within the site), indicating that similar climate and soil conditions support diverse plant communities with diverse stomatal trait assembly. Further, a large part of the variance in stomatal trait moments was not explained by environmental variables. Thus, future studies should consider local factors, such as microclimate, fine-scale soil properties, topography, and even site-specific biotic factors, to improve the predictions of stomatal trait moments at a continental scale.

Reviewer #2 (Remarks to the Author):

Paper by Liu et al., titled “Stomatal morphology-environment relationships of plant communities” is original and novel study greatly fitting for the scope of Nature Communications. It explores community level variability of stomatal morphology (stomatal size and density, stomatal pore index) across multiple forest and grassland ecosystems in China. Authors also explored impact of environmental variables (mean, extreme, seasonality) on variability of stomatal morphology. The scale of the field measurements and plant material sampling is quite impressive and the presented data represent a significant contribution to the scientific community. The results are very interesting and improve our knowledge of environmental drivers of stomatal morphological traits. The paper is overall well written, but English proofreading could help with the stylistic quality of the text and fix minor grammatical errors. Abstract is well composed and highlights the main findings of the paper. Keywords capture the contents of the paper very well. Introduction is quite extensive and provides great overview of problematics for the reader. The materials and methods chapter is well written, with detailed description of study sites, sampling approach, measuring method of stomatal morphology and statistical analyses. The visualizations of results are clear but quality of the figures must be improved. The supplementary materials are also quite extensive, providing further details to reader if needed. Discussion is adequate, but authors could expand on the comparison between forests and grasslands, as well as, include more recent papers focused on the environmental impacts on stomatal morphology. I suggest minor revision of the paper and acceptance in Nature communications, after the authors address my comments:

Response: We are very grateful to the reviewer for the thorough reading of our manuscript and encouraging evaluation, and for the critical comments that have improved the manuscript. All the figures have been improved in this version. Additionally, in our discussion, we have expanded on the comparison between forests and grasslands.

Introduction:

I suggest authors to expand the introduction with focus on significance of stomatal morphology for characterization of the communities. Stomatal morphology (both density and size) affect the drought resistance and WUE of the plants. This highlights the practical importance of this study!

Liu, C., et al., 2018. Variation of stomatal traits from cold temperate to tropical forests and association with water use efficiency. <https://doi.org/10.1111/1365-2435.12973>

Amitrano, C., et al. 2021. Reducing the Evaporative Demand Improves Photosynthesis and Water Use Efficiency of Indoor Cultivated Lettuce. <https://doi.org/10.3390/agronomy11071396>

Stojnić, S., et al. 2019. The use of physiological, biochemical and morpho-anatomical traits in tree breeding for improved water-use efficiency of *Quercus robur* L.

<https://doi.org/10.5424/fs/2019283-15233>

Pitaloka, M.K., et al., 2022. Induced Genetic Variations in Stomatal Density and Size of Rice Strongly Affects Water Use Efficiency and Responses to Drought Stresses.

<https://doi.org/10.3389/fpls.2022.801706>

Response: We greatly appreciated this valuable advice. We have revised the Introduction:

Such as

Line 39–41 *A great number of studies have found that stomatal traits affect **the drought resistance and water use efficiency** of plants, and thus, stomatal traits are frequently and increasingly used in many fields of biology, including ecology and agriculture*

Line 43–46 *analysis of community stomatal traits and their association with environmental variables at a large scale is important for resolving how climate change affects ecosystem functioning, including ecosystem productivity and **water use efficiency** 8, 9*

Line 88–90 *Given that the roles of stomata in **drought resistance and water use efficiency** 3, 29, we hypothesized that environmental variables related to water availability were the main drivers of stomatal traits.*

The references suggested above are now cited in the revised text.

Materials and methods:

Line 114: Above-ground parts were harvested, not was harvest.

Response: Thank you.

*above-ground parts of each species within the plots **were** harvested*

Line 116: Similar to the herb subplots... Or other phrasing.

Response: Here, we rewritten it.

In grasslands, eight plots (1 m × 1 m) in each site were established, then the above-ground biomass of each species was measured using the harvest method

Line 125: There is no shrinkage of the fixed samples in the alcohol solution which could affect the size and density of the stomata?

Response: We have clarified the description. FAA fixative is the most commonly used fixative to fix leaf samples in our field, as it causes minimal shrinkage (as the acetic acid tends to balance the alcohol dehydration)

We revised the text to, “*and were fixed in FAA fixative (75% alcohol: formalin: glacial acetic acid: glycerin)*”

Line:130: Species are limited by labor and expense? Please reformulate the sentences.

Response: Thank you.

Most species in forests are hypostomatic 8, 32, and, given limitation by labor and expense, we only focused on the lower epidermis of forest species.

Line 145: Please include the formula for the SPI calculation. Is it total sample area divided by SD*LA2?

Response: We have added the formula for the SPI calculation

The stomatal pore index (SPI, %) was used to represent the leaf surface covered by stomata

33:

$$SPI = SD \cdot SL^2$$

Results:

Why did you choose to analyse the data only for forest and grassland categories and not for all vegetation types as you are mentioning at line 96: “cold-temperate coniferous forest, temperate deciduous forest, subtropical evergreen forest, tropical rain forest, meadow steppe, typical steppe, and desert steppe”? Ultimately its authors decision how they conceptualize their paper, I am just curious why you chose the simplified design of the data analysis.

Response: We appreciated this question. The vegetation plots were divided into forests and non-forests (in our study, these were grasslands). Although our sample sites included most of the vegetation types in the northern hemisphere, if we used finer vegetation classifications, the sample size would be very small: only 3 sites represent Cold temperate coniferous forest, 2 represent Tropical rain forest, 6 represent Temperate coniferous and broad-leaved mixed forest. Thus, we would lack adequate statistical power to compare these specific types of forests, and we focused on contrasting forests versus grasslands.

Figure 2: The weighted averages for SD ranged from 37 to 456 pores per mm (a), but the weighted variance (d) is ranging up to 100 000 pores per mm ($\log_{10}(100000) = 5$)? Is that correct? Same for the other two parameters.

Response: Yes, this is correct. While at first sight surprising—leading us to carefully checked our dataset several times—we confirmed these numbers. Here is the formula to calculate community-weighted variance:

$$\text{Trait_var} = \sum_1^n p_i (\text{Trait}_i - \text{Trait_mean})^2$$

As the formula for variance contains a square, it results in a large value. For example, if CWM of stomatal density is 200, and the maximum stomatal density within this plot was 800, a large value results: $(800-200)^2 = 360000$.

Generally, a large CWM often corresponded a large CWV, and we plotted the CWM-CWV relationships:

Discussion:

I am missing comparison between the forests and grasslands. Why do the forest communities have larger stomatal density than the grassland communities? Is this driven only by the climate, or also by the prevailing species with different life form (trees vs grasses)?

I suggest authors to expand their discussion and include more recent studies focused on environmental impact of climate on stomatal morphology, e.g.:

Petrík, P., et al. 2020. Stomatal and Leaf Morphology Response of European Beech (*Fagus sylvatica* L.) Provenances Transferred to Contrasting Climatic Conditions.

<https://doi.org/10.3390/fl1121359>

Lin, Y., et al. 2021. Leaf traits from stomata to morphology are associated with climatic and edaphic variables for dominant tropical forest evergreen oaks. <https://doi.org/10.1093/jpe/rtab060>

Sun, J., et al. 2021. Spatial variation of stomatal morphological traits in grassland plants of the Loess Plateau. <https://doi.org/10.1016/j.ecolind.2021.107857>

Petrík, P., et al. 2022. Interannual adjustments in stomatal and leaf morphological traits of European beech (*Fagus sylvatica* L.) demonstrate its climate change acclimation potential.

<https://doi.org/10.1111/plb.13401>

Response: We greatly appreciated this advice, and added a paragraph focused on comparing the forests and grasslands:

Stomatal trait moments varied strongly within and between forests and grasslands. The differences were consistent with theory and empirical findings at species scale, for which higher SD and SPI generally correspond to higher maximum photosynthetic rates and competitiveness, whereas lower SD and SPI would reduce water loss in dry conditions 4, 34, 45, 46. Grassland communities showed lower community weighted means for SD and SPI, which may be an adaptation to conserve water given frequent dry periods in the hot summer growing season, and shallower roots than forest trees; in particular, C4 grass species tend to have lower SD and SPI 47. Further, consistent with our hypothesis that stronger environmental filtering in harsh conditions would result in trait convergence 30, 31, the community-weighted variance of SD and SPI was lower in grasslands than that in forests. This finding indicates that the frequently droughted but competitive environments of grasslands would exclude species with lower and higher SD and SPI values, resulting in trait convergence. The community-weighted kurtosis of SL was lower in grasslands than that in forests, supporting the expectation that stronger environmental filtering would also lead to species being more evenly distributed within the community 48. The community-weighted skewness of SL in forests was higher, consistent with a greater representation of functional rarity 28, associated with extreme trait values for non-dominant species or even rare species, as expected given the environmental and functional heterogeneity of the forest community.

Besides, the references you mentioned above were also cited in the appropriate places.

Line 337: You achieved 32% explanatory power for SD (best climatic parameter). Is it strong enough to be a contradictory result to Bruehlheide et al. 2018? Your findings about environmental drivers of stomatal morphology are great, but still the source for major part of variance is unknown/random as shown in your figure 5.

Response: We appreciated this point and modified that sentence:

In contrast with a previous study of 17 plant traits at the global scale, Bruehlheide, Dengler 20 in which the community-weighted mean and variance were poorly related to environmental variables, we found that stomatal morphology-environment relationships of plant communities strong. Two reasons could explain this discrepancy.

We also added some sentences as follows:

We noted that a large part of the variance in stomatal trait moments was also accounted for by the random site factor (site, plots were nested within the site), indicating that similar climate and soil conditions support diverse plant communities with diverse stomatal trait assembly. Further, a large part of the variance in stomatal trait moments was not explained by environmental variables. Thus, future studies should consider local factors, such as microclimate, fine-scale soil properties, topography, and even site-specific biotic factors, to improve the predictions of stomatal trait moments at a continental scale.

Conclusion:

If you will expand your discussion with comparison between forests and grasslands, it would be fitting to mention that grasslands, which are better adapted to drier conditions have lower stomatal density than forest species.

Response: We have expanded this discussion:

*In conclusion, this first study to reveal stomatal trait relationships of plant communities across broad geographic gradients distinguished strong variation among forest and grassland ecosystems, and trait-trait and trait-climate relationships across communities at continental scale. **We observed differences in stomatal trait moments between forests and grasslands, with lower community-weighted mean and variance of SD and SPI for grasslands than forests;** a trade-off between stomatal density and size both at the species and community scales, suggesting similar constraints by anatomy and environment across scales; and strong influence on community-weighted mean and variance of stomatal traits by environmental variables, with weaker influences on community-weighted skewness and kurtosis. These results highlight the strong patterning of trait–environment relationships across communities at continental scale which will contribute to the power to predict ecosystem productivity in response to future climate change.*

Reviewers' Comments:

Reviewer #1:

Remarks to the Author:

Dear Editor, Dear authors

In this revised version of their manuscript, Congcong Liu and colleagues have dealt satisfactory with some of issues and recommendations of the first review. To my opinion, there are still important issues that the authors need to tackle:

- First, the abstract is not informative enough. The first and last sentences are not specific of the study. Writing that a topic remain unclear and the results are important in regards to climate change could be written for most of research studies. Publication in Nature Communication should be more conclusive.

- The introduction ends with the need of testing relationship between stomatal traits and environmental aridity at community level in order to reveal environmental filtering process. The first result concerns the differences between forest and grassland and the discussion concludes that grassland vs forest differences is a confirmation of environmental filtering evidence. Some pieces of causality are missing to follow author argumentative framework. First, if the first focus is on the differences between grassland and forest, this should be explained in the introduction. I would rather prefer to focus directly on the environmental filtering test, which should occurred similarly in both vegetation types.

- In the Material and Methods, there are approximation in reference and missing information, which is a point underlined in my previous review. There is no information on the method used to determine the tree height. Allometric regressions are not provided. We do not know what is MIPS and the depth of topsoil in soilgrid information. In addition, authors need to check mathematic formulas. Trait means is either formulated 'Mean' or 'Trait_mean' in moment calculation. The calculation of SPI should postulate that stomata were considered as square. The calculation of SPI seems incorrect for amphistomatous species. It was calculated as

$$SPI = \text{mean}(SD_{\text{upp}}, SD_{\text{low}}) * (SL_{\text{upp}} + SL_{\text{low}})^2$$

This should rather be $SPI = (SD_{\text{upp}} * SL_{\text{upp}}^2) + (SD_{\text{low}} * SL_{\text{low}}^2)$

This could also be $SPI = (SD_{\text{upp}} * SL_{\text{upp}} * SW_{\text{upp}}) + (SD_{\text{low}} * SL_{\text{low}} * SW_{\text{low}})$, if we consider that stomata are rectangular rather than square

Reference 33 is very correct to justify the difference in measurement and calculation for forest vs grassland species, however reference 8 is not.

Finally, in the statistical analysis section, it is not explained how the different analyses are related to the objectives of the study. For instance, the PCA should be justified or removed as it is not clear how this is related to the test of environmental filtering at the community scale.

Overall, I recommend a more sound verification of the method section.

- In the result, figure 3 should be log-transformed to better visualize the trade-off. I recommend a better justification for Figures 2 and 4 in regards to the main objective.

Hope that the recommendation will be useful.

Reviewer #2:

Remarks to the Author:

The study presents large-scale analysis of community weighted stomatal morphological acclimation to environmental conditions across China. Authors refined the quality of text and added missing details to introduction, materials and methods, discussion and conclusion. The newly revised version reads well and gives clear comprehensive picture of the results. Based on the revised manuscript and response letter I can conclude that the authors addressed all my comments sufficiently.

I am suggesting the editors acceptance of the manuscript for the publication in Nature Communications.

Reviewer #1 (Remarks to the Author):

Dear Prof. Maire,

Thank you for approving our latest revisions, and your valuable comments indeed helped us improve our manuscript.

- First, the abstract is not informative enough. The first and last sentences are not specific of the study. Writing that a topic remain unclear and the results are important in regards to climate change could be written for most of research studies. Publication in Nature Communication should be more conclusive.

Response: We agree, and have revised the Abstract.

The first sentence was changed to: *The association of stomatal traits with environmental drivers across communities has important implications for ecosystem fluxes but have remained unclear.*

The last sentence was changed to: *Our findings extend the knowledge of stomatal trait–environment relationships to the ecosystem scale, with applications in predicting future water and carbon cycles.*

- The introduction ends with the need of testing relationship between stomatal traits and environmental aridity at community level in order to reveal environmental filtering process. The first result concerns the differences between forest and grassland and the discussion concludes that grassland vs forest differences is a confirmation of environmental filtering evidence. Some pieces of causality are missing to follow author argumentative framework. First, if the first focus is on the differences between grassland and forest, this should be explained in the introduction. I would rather prefer to focus directly on the environmental filtering test, which should occurred similarly in both vegetation types.

Response: We appreciated these points. To address them, first, we added some sentences in the Introduction to mention the differences between forests and grasslands. Line 104-112.

We distinguished forests and grasslands, given their major divergence in life form composition and ecosystem structure, and the greater environmental stress typically experienced by grasslands, such as severe drought, and, additionally, the higher water use efficiency of C4 grasses. Given that one of the foundations of trait-based ecology is that trait-environment relationships tend to be consistent 33, we tested whether stomatal trait-environment relationships were consistent in forests and grasslands. We hypothesized that grasslands would tend to show greater convergence toward conservative stomatal traits, i.e., lower stomatal density and a smaller proportion of the leaf epidermis allocated to stomata, to reduce maximum rates of transpiration and improve water use efficiency.

Second, we also added an analysis to test the environmental filtering in grasslands and forests. We have added some sentences in the Methods section to explain how we tested the environmental

filtering, and in the Results we report the novel findings of strong environmental filtering within these ecosystem types.

Method Section

We tested for environmental filtering could as the deviation of trait moments from random expectation. We randomized stomatal trait values across all species 500 times. For each run we calculated stomatal trait moments with the randomized trait values, but retaining the species set and their abundances in each plant community intact 21. For each stomatal trait, we calculated standardized effect sizes (SESs) for the variance in CWMs and for the mean in CWVs, CWSs, and CWKs. SES was calculated as

$$SES_{var(CWM)} = \frac{\text{var}(CWM_{obs}) - \text{mean}(\text{var}(CWM_{ran}))}{s.d.(\text{var}(CWM_{ran}))}$$

$$SES_{mean(CWV, or CWS, or CWK)} = \frac{\text{mean}(CWV_{obs}, or CWS_{obs}, or CWK_{obs}) - \text{mean}(CWV_{ran}, or CWS_{ran}, or CWK_{ran})}{s.d.(\text{mean}(CWV_{ran}, or CWS_{ran}, or CWK_{ran}))}$$

where var, mean, and s.d. represented computing the mean, variance, and standard deviation of the vector. Positive SESs for the variance of CWM and negative SESs for the means of CWV, CWS, and CWK would be consistent with environmental filtering of stomatal traits 21. These processes were also performed separately for forests and grassland.

Results Section

Standardized effect sizes of stomatal trait moments

Overall, tests of standardized effect sizes (SESs) for stomatal moments indicated strong environmental filtering. Considering forests and grasslands together, the SESs for the variance in CWMs were positive, and those for the means of CWVs, CWSs, and CWKs were negative (Table S2), and all of them were consistent with the predictions of the environmental filtering hypothesis. When considering only forests, the SESs for the variance in CWMs and for the means of CWSs and CWKs were consistent with the predictions of the environmental filtering hypothesis. When considering only grasslands, the SESs for the means of CWVs, CWSs, and CWKs were consistent with the predictions of the environmental filtering hypothesis.

Table S2 Standardized effect sizes (SESs) for the variance in CWMs and for the mean in CWVs, CWSs, and CWKs.

	Trait moments	Expected	All	Forests	Grasslands
SES _{var}	SD_mean	+	2.62	6.78	-3.21
SES _{var}	SL_mean	+	3.93	4.98	2.97
SES _{var}	SPI_mean	+	3.31	6.61	-0.77
SES _{mean}	SD_var	-	-1.50	17.31	-8.93
SES _{mean}	SL_var	-	-0.30	2.76	-1.83

SES _{mean}	SPI_var	-	-0.03	16.07	-7.21
SES _{mean}	SD_skew	-	-1.22	0.07	-1.76
SES _{mean}	SL_skew	-	-3.50	-0.37	-5.03
SES _{mean}	SPI_skew	-	-1.62	-0.65	-2.21
SES _{mean}	SD_kurt	-	-1.77	-2.18	-1.62
SES _{mean}	SL_kurt	-	-2.05	-2.29	-1.95
SES _{mean}	SPI_kurt	-	-1.83	-1.61	-2.04

Discussion Section

Across all plant communities, the observed variances in CWMs of the stomatal traits were higher than expected by chance (all SESs >0), and the observed means in CWVs, CWSs, and CWKs of these stomatal traits were lower than expected by chance (all SESs <0). These findings indicate that environmental filtering influenced community stomatal trait composition on a large scale. When forests and grasslands were analyzed separately, the means of CWSs and CWKs of both forests and grasslands were consistent with the predictions of the environmental filtering hypothesis, whereas the means of CWVs of forests and the variances of CWMs of grasslands did not reflect environmental filtering. A possible explanation for these patterns is that CWS and CWK were mainly influenced by rare phenotypes, which are likely to be filtered out of a community by environmental stress 32. Therefore, compared with CWM and CWV, CWS and CWK showed a more robust support for environmental filtering, especially in the narrower environmental gradients for each ecosystem type. This coincided with previous studies that using CWS and CWK to unveil ecological assembly rules 39.

- In the Material and Methods, there are approximation in reference and missing information, which is a point underlined in my previous review. There is no information on the method used to determine the tree height. Allometric regressions are not provided. We do not know what is MIPS and the depth of topsoil in soilgrid information.

Response:

In this revision, we have added clarification for how we measured the tree height

The height and diameter at breast height (DBH) of each individual tree was recorded. Specifically, the height of the small trees was measured using telescopic rods, and the height of the large trees was measured with the Blume-Leiss hypsometer or directly in permanent sample plots with a tower crane.

For allometric regressions, we obtained from the books

《Carbon storage in forest ecosystems in China- biomass models》 ISBN: 9787508853925,

《Handbook of biomass models of common shrubs in China》 ISBN: 9787508854380

We could easily obtain the allometric regressions from the books. Unfortunately, the two books are only available in Chinese, not in English. The titles of the books were also translated by ourselves, not the authors.

The following link gives a preview:

<https://book.sciencereading.cn/shop/book/Booksimple/onlineRead.do?id=B6E9625F159E0316AE053020B0A0AC4D3000&bookPageNum=6>

Using these allometric regressions, the carbon storage of China was estimated by the authors of this

book, and their paper were published in PNAS, titled “Carbon pools in China’s terrestrial ecosystems: New estimates based on an intensive field survey”

We also added some sentences in the MS,

The organs and total biomass of individual trees and shrubs were calculated using species-specific allometric regressions based on the measured values of DBH or basal diameter and height. Species-specific allometric regressions were obtained from the books, “Carbon storage in forest ecosystems in China: biomass models”, and “Handbook of biomass models of common shrubs in China”. When the allometric regression of specific species was not available, we used the allometric regression of the same genera, or mixed-species equations of a forest.

MIPS is an electronic image analysis software with functionality similar to Image J. We added further description to the Methods. Here is the operation interface of MIPS.

the depth of topsoil in soilgrid information was 0-5cm, the sentence was re-written as follows:

Key physico-chemical properties (bulk density, soil total nitrogen, soil pH, and sand-silt-clay content) of topsoil (0-5cm depth) were collected from the SoilGrids

In addition, authors need to check mathematic formulas. Trait means is either formulated ‘Mean’ or ‘Trait_mean’ in moment calculation.

Response: Thank you for your correction. We have revised it as follows:

$$\begin{aligned}\text{Trait_mean} &= \sum_1^n p_i \text{Trait}_i \\ \text{Trait_var} &= \sum_1^n p_i (\text{Trait}_i - \text{Trait_mean})^2 \\ \text{Trait_skew} &= \sum_1^n \frac{p_i (\text{Trait}_i - \text{Trait_mean})^3}{\text{Trait_var}^{\frac{3}{2}}}\end{aligned}$$

$$\text{Trait_kurt} = \sum_1^n \frac{p_i(\text{Trait}_i - \text{Trait_mean})^4}{\text{Trait_var}^2}$$

The calculation of SPI should postulate that stomata were considered as square. The calculation of SPI seems incorrect for amphistomatous species. It was calculated as

$$\text{SPI} = \text{mean}(\text{SD}_{\text{upp}}, \text{SD}_{\text{low}}) * (\text{SL}_{\text{upp}} + \text{SL}_{\text{low}})^2$$

$$\text{This should rather be SPI} = (\text{SD}_{\text{upp}} * \text{SL}_{\text{upp}}^2) + (\text{SD}_{\text{low}} * \text{SL}_{\text{low}}^2)$$

This could also be $\text{SPI} = (\text{SD}_{\text{upp}} * \text{SL}_{\text{upp}} * \text{SW}_{\text{upp}}) + (\text{SD}_{\text{low}} * \text{SL}_{\text{low}} * \text{SW}_{\text{low}})$, if we consider that stomata are rectangular rather than square

Response:

We are grateful to the reviewer for highlighting this point to clarify. SD and SL are the most commonly measured stomatal traits, and these were the focus of the study. For amphistomatous species, their SD was the sum of the SD_{upp} and SD_{low} , their SL was the average of SL_{upp} and SL_{low} .

We have revised our sentences as follows:

For amphistomatous species, SD was the sum of the upper and lower stomatal density, and SL was the average of the upper and lower stomatal length.

Stomatal size was very similar across leaf surfaces ($y = 0.999x$, $R^2 = 0.994$).

Given that SL was similar in lower and upper leaf surface, we calculated SPI as $\text{SPI} = \text{SD} * \text{SL}^2$
SD was the sum of the upper and lower stomatal density, and SL was the average of the upper and lower stomatal length.

We tested this method against the equation suggested by the reviewer for SPI

$$\text{SPI} = (\text{SD}_{\text{upp}} + \text{SD}_{\text{low}}) * ((\text{SL}_{\text{upp}} + \text{SL}_{\text{low}})/2)^2 = \text{SD} * \text{SL}^2 \text{ eqn1}$$

$$\text{SPI} = (\text{SD}_{\text{upp}} * \text{SL}_{\text{upp}}^2) + (\text{SD}_{\text{low}} * \text{SL}_{\text{low}}^2) \text{ eqn2}$$

As we can see from the figure, there was no significant difference between the two calculation methods.

$SPI = SD * SL^2$ was created by Lawren Sack, it has been a widely accepted stomatal trait in our field, This terminology SPI was first appeared in a manuscript titled “The ‘hydrology’ of leaves: coordination of structure and function in temperate woody species.” which has been cited more than 800 times.

As we can see from the equation $SPI = SD * SL^2$, stomata were considered as square (or, SPI would be proportionally related to pore area if this is a disk, as SL^2 would simply be multiplied by $\pi/4$). We also could consider that the stomatal pore is rectangular, and calculate of SPI as

$$SPI = (SD_{upp} * SL_{upp} * SW_{upp}) + (SD_{low} * SL_{low} * SW_{low}) \quad \text{eqn 3}$$

In fact, stomatal length (SL) and stomatal width (SW) were positively correlated. We also measured the SW in our study.

We also found that SPI-eqn 1 and SPI-eqn3 were also strongly correlated.

Therefore, this method of calculating SPI is reliable.

In summary,

- 1) SD and SL were the most important stomatal traits, and using SD and SL enables direct calculation of SPI
- 2) $SPI = SD * SL^2$ has been widely accepted as a comparative index of stomatal pore area, it has been used many hundred times in our field.
- 3) If we considered the SW in the SPI (which would probably overwhelm readers with too many stomatal traits), the results would be the same, as SPI-eqn1 was strongly correlated with SPI-eqn3 ($R^2 = 0.98$). Thus our conclusions are robust.

Reference 33 is very correct to justify the difference in measurement and calculation for forest vs grassland species, however reference 8 is not.

Response:

Thank you for correcting our mistake,

Finally, in the statistical analysis section, it is not explained how the different analyses are related to the objectives of the study. For instance, the PCA should be justified or removed as it is not clear how this is related to the test of environmental filtering at the community scale.

Response:

We agree that the PCA was not related to our main objective. Following your advice, we removed it from the Methods and Results Section.

Overall, I recommend a more sound verification of the method section.

Response:

Following your above advice, we have revised the method section carefully. Thank you for your help.

- In the result, figure 3 should be log-transformed to better visualize the trade-off. I recommend a better justification for Figures 2 and 4 in regards to the main objective.

Response:

We agreed with you, and now the axes for stomatal density and size were log-transformed. The Figure 3 was redrawn as follows:

We also have added some sentences in the Introduction to explain why the comparisons between grassland and forests were needed, and why the stomatal trait-environment relationships were explored separately in forests and grasslands.

We distinguished forests and grasslands, given their major divergence in life form composition and ecosystem structure, and the greater environmental stress typically experienced by grasslands, such as severe drought, and, additionally, the higher water use efficiency of C4 grasses. Given that one of the foundations of trait-based ecology is that trait-environment relationships tend to be consistent 33, we tested whether stomatal trait-environment relationships were consistent in forests and grasslands. We hypothesized that grasslands would tend to show greater convergence toward conservative stomatal traits, i.e., lower stomatal density and a smaller proportion of the leaf epidermis allocated to stomata, to reduce maximum rates of transpiration and improve water use efficiency.

Hope that the recommendation will be useful.

Response: We have carefully considered these comments, and all of us agree that these greatly improved our article. Thank you very much for your valuable help.

Sincerely,

Congcong Liu and Nianpeng He

Reviewer #2 (Remarks to the Author):

The study presents large-scale analysis of community weighted stomatal morphological acclimation to environmental conditions across China. Authors refined the quality of text and added missing details to introduction, materials and methods, discussion and conclusion. The newly revised version reads well and gives clear comprehensive picture of the results. Based on the revised manuscript and response letter I can conclude that the authors addressed all my comments sufficiently. I am suggesting the editors acceptance of the manuscript for the publication in Nature Communications.

Response: Thanks for the positive review. With your assistance, we are confident that this manuscript will have a significant impact in our field.